# Controlling diverse robots by inferring Jacobian fields with deep networks

Sizhe Lester Li[1✉], Annan Zhang[1], Boyuan Chen[1], Hanna Matusik[1], Chao Liu[1], Daniela Rus[1] & Vincent Sitzmann[1✉]

Mirroring the complex structures and diverse functions of natural organisms is a long-standing challenge in robotics[1–4]. Modern fabrication techniques have greatly expanded the feasible hardware[5–8], but using these systems requires control software to translate the desired motions into actuator commands. Conventional robots can easily be modelled as rigid links connected by joints, but it remains an open challenge to model and control biologically inspired robots that are often soft or made of several materials, lack sensing capabilities and may change their material properties with use[9–12]. Here, we introduce a method that uses deep neural networks to map a video stream of a robot to its visuomotor Jacobian field (the sensitivity of all 3D points to the robot's actuators). Our method enables the control of robots from only a single camera, makes no assumptions about the robots' materials, actuation or sensing, and is trained without expert intervention by observing the execution of random commands. We demonstrate our method on a diverse set of robot manipulators that vary in actuation, materials, fabrication and cost. Our approach achieves accurate closed-loop control and recovers the causal dynamic structure of each robot. Because it enables robot control using a generic camera as the only sensor, we anticipate that our work will broaden the design space of robotic systems and serve as a starting point for lowering the barrier to robotic automation.

Modern manufacturing techniques promise a new generation of robotic systems inspired by the diverse mechanisms seen in nature. Whereas conventional systems are precision engineered from rigid parts connected at discrete joints, biologically inspired robots generally combine soft, compliant materials and rigid parts, and often forgo conventional motor-driven actuation for pneumatic and muscle-like actuators[9]. Previous work has demonstrated that such hybrid soft–rigid systems can already outperform conventional counterparts in certain environments in which adaptation to changing circumstances[13,14] or safety in co-working with humans is key[15]. Furthermore, these systems are amenable to mass production, some requiring no human assembly[6], and may thus substantially lower the cost and barriers to robotic automation[16]. However, the use of bio-inspired hardware is hindered by our limited capability to model these systems, because any robotic system needs to be paired with a model that can accurately predict the motion of key components, such as the end-effector, under all possible commands at all times.

Conventional robots were designed to make their modelling and control easy. They are usually constructed from precision-machined parts fabricated from high-stiffness materials with Young's moduli in the range $10^9–10^{12}$ Pa (ref. 9). Connected by low-tolerance joints, these rigid robots are adequately modelled as a kinematic chain consisting of idealized rigid links. Accurate sensors in every joint then allow a faithful 3D reconstruction of the robot during its use. With this in place, an expert can reliably model the motion of the robot under all possible motor commands and design control algorithms to execute desired motions.

By contrast, the bodies of soft and bio-inspired robots are difficult to model. They are typically made of materials that match the stiffness of soft biological materials such as tissue, muscles or tendons[5,9,17]. These materials undergo large deformations during actuation and exhibit time-dependent effects, such as viscoelasticity and gradual weakening through repeated loading and unloading. Partial differential equations that govern the behaviour of soft materials derived from continuum mechanics and large deformation theory are costly to solve, especially for control and real-time applications. Model order reduction methods, geometrical approximation methods and rigid discretization methods rely heavily on simplifying assumptions about the specific system and do not universally generalize[10–12,18]. Previous work has leveraged machine learning[19–23] and marker-based visual servoing[24–27] to overcome these challenges, but they require extensive expert-guided customization to be applied to a particular robot architecture. Furthermore, high-precision motion-capture systems (for example, OptiTrack, VICON and Qualisys) are costly, bulky and require a controlled setting for their use. Other work has explored neural scene representations of robot morphology[28], but this assumes precise embedded sensors not available in soft and bio-inspired robots and relies on 3D motion capture for fine-grained control. What is required is a general-purpose control method that is agnostic to the fabrication, actuation, embedded sensors, material and morphology of the robotic system.

The work in this article was inspired by human perception. Controlling robots with a video-game controller, people can learn to pick and place objects within minutes[29]. The only sensors we require are our

[1]Computer Science and Artificial Intelligence Laboratory, Massachusetts Institute of Technology, Cambridge, MA, USA. ✉e-mail: sizheli@mit.edu; sitzmann@mit.edu

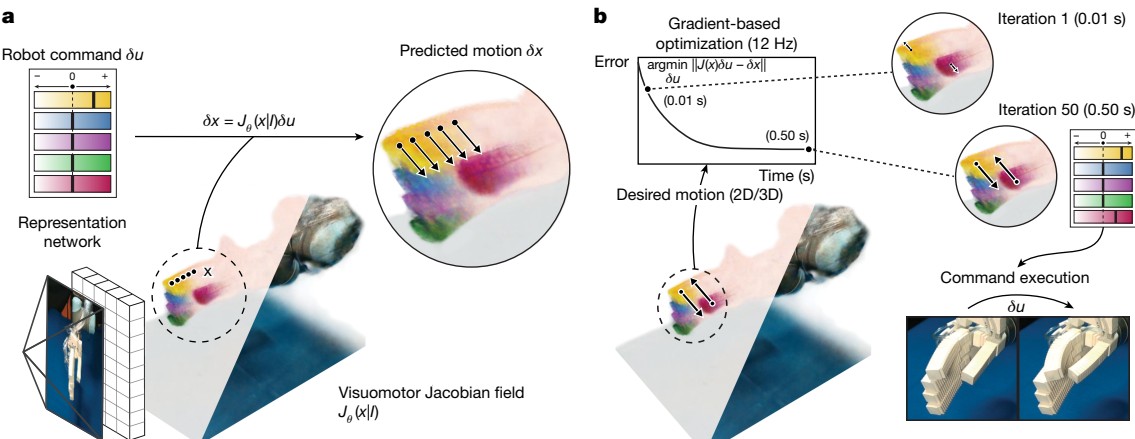

**Fig. 1 | Controlling robots from vision using the visuomotor Jacobian field. a**, Reconstruction of the visuomotor Jacobian field and motion prediction. From a single image, a machine-learning model infers a 3D representation of the robot in the scene, which we name the visuomotor Jacobian field. It encodes the robot's geometry and kinematics, enabling us to predict the 3D motions of robot surface points under all possible commands. Colours indicate the sensitivity of that point to individual command channels. **b**, Closed-loop control from vision. Given desired motion trajectories in pixel space or in 3D, we use the visuomotor Jacobian field to optimize for the robot command that would generate the prescribed motion at an interactive speed of approximately 12 Hz. Executing the robot command in the real world confirms that the desired motions have been achieved.

eyes. From vision alone, we can learn to reconstruct the robot's 3D configuration and to predict its motion as a function of the control inputs we generate.

In this article, we introduce visuomotor Jacobian fields, a machine-learning approach that can control robots from a single video-camera stream. We trained our framework using 2–3 hours of multi-view videos of the robot executing randomly generated commands captured by 12 consumer-grade RGB-D video cameras. No human annotation or expert customization is necessary to learn to control a new robot. After training, our method can control the robot to execute the desired motions using only a single video camera. Relying on vision as the only sensor, visuomotor Jacobian fields do not make assumptions about the kinematics, dynamics, material, actuation or sensing capabilities of the robot. Our method is uniquely enabled by recent advancements in computer vision, neural scene representation and motion tracking.

We evaluate visuomotor Jacobian fields on a wide range of robotic manipulation systems, specifically a 3D-printed hybrid soft–rigid pneumatic hand[30], a compliant wrist-like robotic platform made of handed shearing auxetics (HSAs)[31], a commercially available Allegro Hand with 16 degrees of freedom, and a low-cost educational-robot arm[32]. Across all these systems, we show that our method reliably learns to reconstruct their 3D configuration and predict their motion at all times.

Our method unshackles the hardware design of robots from our ability to model them manually, which in the past has dictated precision manufacturing, costly materials, extensive sensing capabilities and reliance on conventional, rigid building blocks. Our method therefore has the potential to substantially broaden the design space of robots that can be deployed in practice, as well as lowering the cost and barriers to adopting robotic automation by enabling the precision control of low-cost robots.

## The visuomotor Jacobian field

Our framework comprises two key components: first, a deep-learning-based state-estimation model that infers a 3D representation of the robot that encodes both its 3D geometry and its differential kinematics—how any point in 3D will move under any possible robot command—from only a single video stream; and second, an inverse dynamics controller that parameterizes desired motions densely in the 2D image space or 3D, and finds robot commands at interactive speeds. We found that parameterizing demonstration trajectories as

dense point motions is the key to controlling a diverse range of robotic systems, because the motions of deformable and dexterous robots cannot be well constrained by rigid transformations specified on a single 3D frame. Our parameterization enables a wide range of systems to imitate video-based demonstrations. A schematic overview of the system we used is shown in Fig. 1.

The state-estimation model is a deep-learning architecture that maps a single image, $\mathbf{I}$, of the robot to a 3D neural scene representation. This 3D representation maps any 3D coordinate to features that describe the robot's geometric and kinematic properties at that 3D coordinate[33–35]. Specifically, we reconstruct both a neural radiance field[33], which encodes the robot's 3D shape and appearance at every 3D coordinate, and an innovative visuomotor Jacobian field, which maps each point in 3D to a linear operator that expresses that point's 3D motion as a function of robot actuator commands. The neural radiance field maps a 3D coordinate to its density and radiance, which serves as a representation of the geometry of the robot.

The visuomotor Jacobian field encodes how any 3D coordinate will move as a function of any possible actuator command. It serves as a representation of the differential kinematics of the robot and generalizes the conventional system Jacobian in robotics. Robots are conventionally modelled by designing a dynamical system that has state $\mathbf{q} \in \mathbb{R}^m$, input command $\mathbf{u} \in \mathbb{R}^n$ and dynamics $\mathbf{q}^+ = \mathbf{f}(\mathbf{q}, \mathbf{u})$, where $\mathbf{q}^+$ denotes the state of the next time step. The system Jacobian $\mathbf{J}(\mathbf{q}, \mathbf{u}) = \partial \mathbf{f}(\mathbf{q}, \mathbf{u}) / \partial \mathbf{u}$ is the matrix that relates the change of command $\mathbf{u}$ to the change of state $\mathbf{q}$, which arises from the linearization of $\mathbf{f}$ around the nominal point $(\bar{\mathbf{q}}, \bar{\mathbf{u}})$, as $\delta \mathbf{q} = \mathbf{J}|_{\bar{\mathbf{q}}, \bar{\mathbf{u}}} \delta \mathbf{u}$. This approach relies on experts to design the system's state encoding $\mathbf{q}$ and dynamics $\mathbf{f}$ on a case-by-case basis. Although this is feasible for conventional robots, it is challenging for hybrid soft–rigid, insufficiently sensorized and under-actuated systems, or systems with significant backlash, that is, play or 'wiggle' caused by loose tolerances of joints and gears resulting from imprecise manufacturing[36,37].

Our visuomotor Jacobian field instead directly maps any 3D point $\mathbf{x}$ to its corresponding system Jacobian. Instead of conditioning on an expert-designed state representation $\mathbf{q}$, the Jacobian field is reconstructed directly from the input image $\mathbf{I}$ by deep learning[38]. As shown in the Supplementary Information, our 3D Jacobian parameterization injects linearity and spatial locality inductive biases that are essential for generalization to unseen robot configurations and motor commands. Specifically, $\mathbf{J}(\mathbf{x}, \mathbf{I}) = \partial \mathbf{x} / \partial \mathbf{u}$ describes how a change of actuator

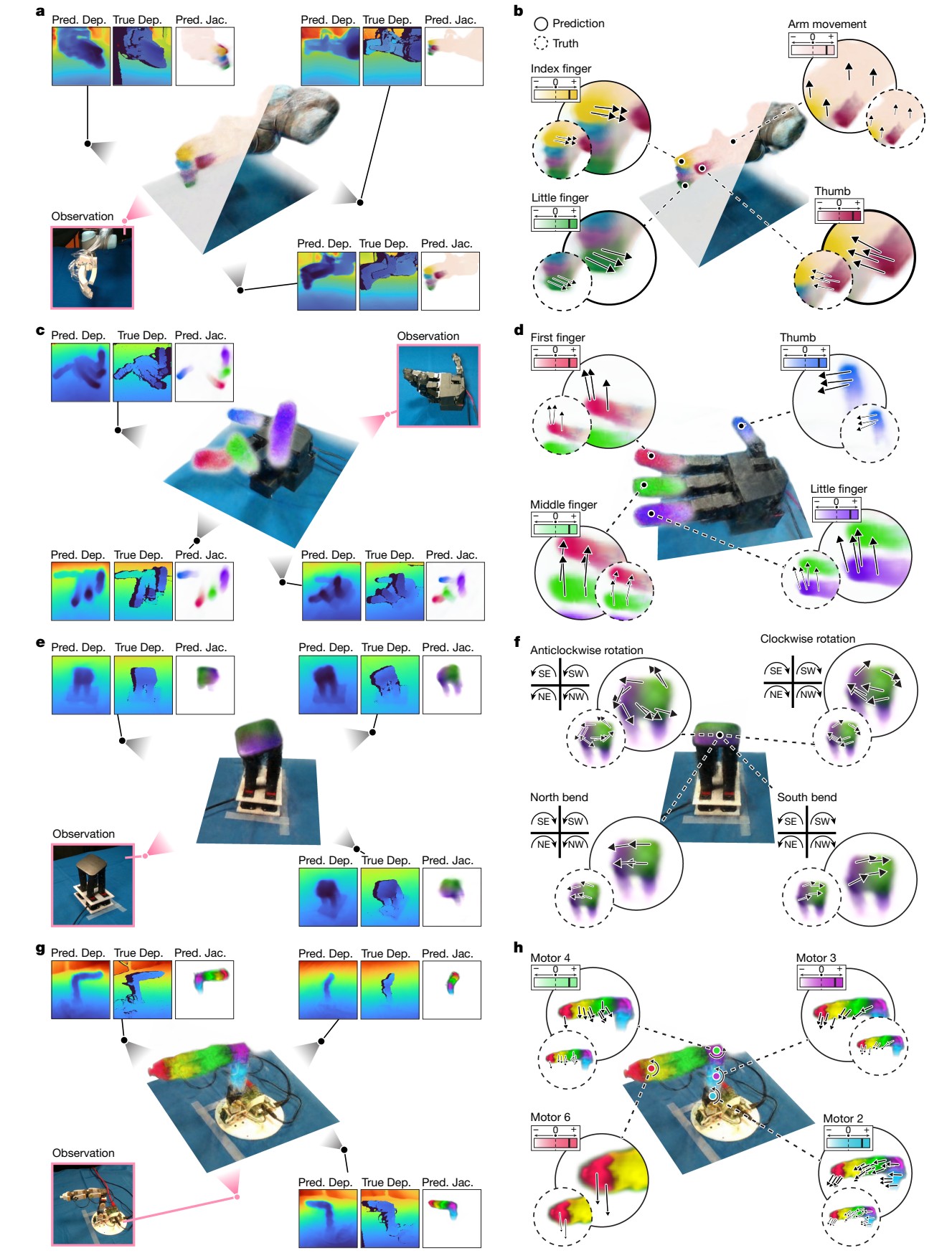

**Fig. 2** | See next page for caption.

state $\delta\mathbf{u}$ relates to the change of 3D motion at coordinate $\mathbf{x}$, through $\delta\mathbf{x} = \mathbf{J}(\mathbf{x}, \mathbf{I})\delta\mathbf{u}$. This allows us to densely predict the 3D motion of any point in space using $\delta\mathbf{x} = \mathbf{J}(\mathbf{x}, \mathbf{I})\delta\mathbf{u}$. Figure 1a and Fig. 2b,d,f,h illustrate the 3D motions predicted by our Jacobian field across a range of robot platforms.

To reconstruct visuomotor Jacobian and radiance fields, we rely on a neural single-view-to-3D module[38]. By reconstructing both robot geometry and kinematics directly from a camera observation, our state estimation model is agnostic to the sensors embedded in the robot. Instead of expert-modelling the relationship between motor and sensor readings and the 3D geometry and kinematics of the robot, our system learns to regress this relationship from data directly.

Our state estimation model was trained in a self-supervised way using video streams from 12 RGB-D cameras that observed the robot executing random commands from different perspectives. We provide a detailed illustration of the training process in Extended Data Fig. 1. For each camera stream, we extracted 2D motion using optical flow and point-tracking methods. At every training step, we selected one of the 12 cameras as input for our reconstruction method. From this single input image, we reconstructed the visuomotor Jacobian and radiance fields that encode the robot's 3D geometry and appearance. Given a robot command, we used the Jacobian field to predict the resulting 3D motion field. We used volume rendering[33] to render the 3D motion field to the 2D optical flow of one of the other 12 cameras and compared it with the observed optical flow. This procedure trains the Jacobian field to predict robot motion accurately. We further volume rendered the radiance field from one of the 12 cameras and compared the RGB and depth outputs with the captured RGB-D images, which trained our model to reconstruct accurate 3D geometry. Differentiable rendering[35,39,40] is a convenient and effective source of 3D supervision, but alternative sources of 3D supervision could be equally effective, as discussed in the Supplementary Information.

When deployed, our framework can solve for commands that implement a given 3D or 2D image-space motion trajectory, enabling two applications in robot manipulation. The ability to translate 2D image-space motions into robot commands is meaningful for general imitation learning. As validated in Figs. 3 and 4, tracking 2D image-space trajectories enables robot manipulation in the face of changing robot dynamics and scene appearances resulting from factors such as external loads, physical wear and tear and occlusions. To solve for a robot command, we used Jacobian fields to solve the optimization problem (Fig. 1b) to find the best robot command that produces the desired 2D motion. We translated robot videos to reference trajectories by extracting robot surface point tracks. As described in the Supplementary Information, a simple model predictive control (MPC) algorithm achieves closed-loop trajectory tracking.

For the 3D application, our framework enables demonstration transfer between viewpoints. A video recorded from a viewpoint that is no longer available at deployment time can still be used for trajectory following. Our model achieves this by lifting each 2D video frame to a 3D particle state, translating the video to a 3D trajectory. Given the new camera viewpoint at deployment time, our model lifts the current observation into the 3D particle state that is consistent with the 3D demonstration trajectory. We achieve 3D trajectory tracking using the same MPC algorithm and changing only the cost function to 3D shape distance, as described in the Supplementary Information.

We demonstrated the capabilities enabled by our framework by controlling robotic systems that covered diverse types of material, varying kinematic complexity and different price points. In summary, we controlled a US$300 3D-printed hybrid soft–rigid pneumatic hand mounted on a conventional robot arm, a soft parallel manipulator made from handed shearing auxetics, a rigid Allegro hand with 16 degrees of freedom, and a manually assembled DIY robot arm with 3D-printed parts, low-cost motors and substantial backlash[37,41].

As shown in Fig. 2, for each of these challenging robotic systems, visuomotor Jacobian fields succeeded in reconstructing an accurate 3D representation of the respective robot from just a single image (Supplementary Video 1). We assigned a unique colour to the influence of each channel of the $n$-dimensional motor command and visualized the Jacobian field. We found that our Jacobian field learnt the causal kinematic structure of each robot, identifying which command channel was responsible for actuating which part of the robot in 3D space. This capability arose fully self-supervised, without any annotation or supervision that would match motors with robot parts. We qualitatively found that 3D motions predicted by our framework given robot commands highly agreed with the ground truth reference motions (Fig. 2). Quantitatively, our method reconstructed high-quality geometry and dynamics from just a single RGB input view across robotic systems (Extended Data Table 1). For perception, the mean depth-prediction error was largest for the pneumatic hand (6.519 mm) and smallest for the HSA platform (1.109 mm). The pneumatic hand was actuated through translucent tubing that could pose challenges to geometry prediction. For flow prediction, our framework achieved mean prediction errors of 1.305 pixels and 1.150 pixels on the Allegro hand and pneumatic hand, respectively. The Poppy robot arm had the most significant mean flow prediction error of 6.797 pixels, because the hardware constantly experienced backlash resulting from low-quality hardware. Our method could model the rotational and bending mechanisms of the HSA platform and achieved a mean flow prediction error of 3.597 pixels. Furthermore, we found that predicted Jacobians closely matched the references computed by expert-crafted kinematic models (Fig. 4f, Extended Data Fig. 6 and Extended Data Table 4). In Extended Data Fig. 5, Extended Data Table 3 and Supplementary Information, we benchmark Jacobian fields with a pure neural-network-based 3D scene flow prediction method in real-world and simulated environments, demonstrating that our method is key to generalizing to unseen robot motion and motor commands at test time. In Extended Data Fig. 3, we demonstrate our system's robustness against visual occlusions.

We next evaluated the performance of our method for closed-loop control. For the Allegro hand, we prescribed the controller with a 2D trajectory that tracked a desired pose (Fig. 3b). On completion of the trajectory, we quantified error using the built-in, high-precision per-joint sensors and the hand's precise 3D forward kinematics model. Purely from vision, our system controlled the Allegro hand to close and open every finger fully, achieving errors of less than 3° per joint and less than 4 mm for each fingertip (Fig. 4d, Extended Data Table 1 and Extended Data Fig. 2). We demonstrated on the HSA platform that our system can successfully control robots under substantially changed dynamics without retraining. We intentionally disturbed the

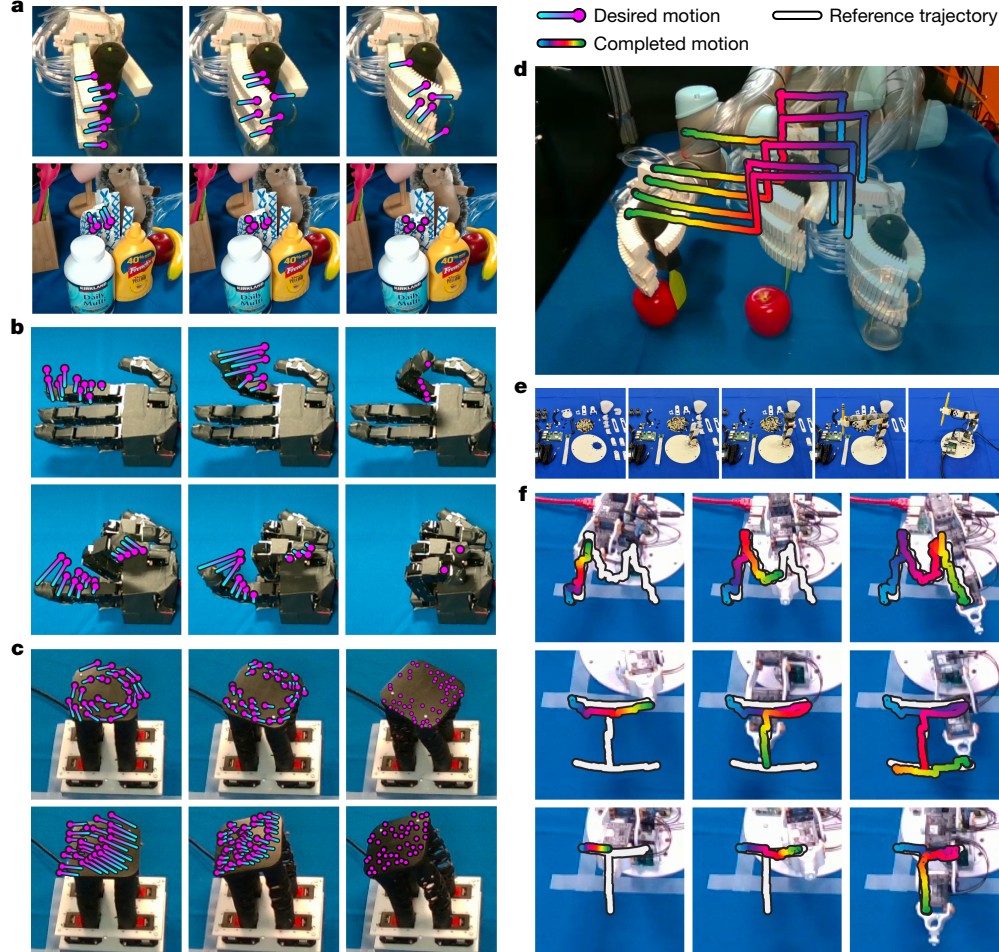

**Fig. 3 | Closed-loop control of diverse robots from vision. a**, We controlled a 3D-printed soft–rigid pneumatic hand to complete a grasp (top) and execute finger motion under the presence of occluders (bottom). **b**, Our method precisely controlled every finger of the Allegro hand to close and form a fist. **c**, We used our system to control complex rotational and bending motions on a wrist-like, soft-handed shearing auxetics platform. **d**, We controlled a soft–rigid pneumatic hand mounted on a UR5 robot arm to accomplish a tool grasp and a pushing action. **e**, The process of assembling a low-cost, 3D-printed robot arm that is difficult to model and lacks sensing. **f**, Our method was shown to be robust against backlash, that is, play or 'wiggle' caused by loose tolerances of joints and gears of the Poppy robot described in ref. 41, enabling the robot arm to draw the letters M, I, and T in the air. These motion sequences were out of distribution and were not part of our training data.

HSA platform by attaching calibration weights with a total mass of 350 g to a wooden rod, which we glued to the top of the HSA platform. The weights exerted a vertical force and a torque on the platform top, which made it tilt visibly in its resting position (Fig. 4a). Furthermore, the rod and the weights constituted a visual disturbance. We used the OptiTrack motion-capture system, which had less than 0.2 mm of measurement error, and attached markers on the surface of the HSA platform to quantify the position errors in goal-pose tracking. We found that our vision-based framework was capable of controlling the robot to complete complex rotational motions and reach the target configuration, achieving an error of 7.303 mm, effectively overcoming the external perturbation on the system's dynamics (Fig. 4b,c and Extended Data Table 1). For the 3D-printed Poppy robot arm, we designed target trajectories demanding the robot to draw a square and the letters MIT in the air. These motion sequences were out of distribution and were not in our training data. We attached OptiTrack markers on the end-effector of the robot arm to measure errors in 3D position. Our framework achieved an average error of less than 6 mm in the goal-pose tracking task (Fig. 4e and Extended Data Table 1).

We also evaluated our model's ability to enable demonstration transfer between viewpoints. Our model achieved this by lifting each 2D video frame to a 3D particle state, translating the video to a 3D trajectory. Quantitatively, our method achieved a low median error of 2.2° (Extended Data Fig. 4c).

Overall, our framework enabled precise control of diverse robotic systems, including both conventional rigid systems and 3D-printed hybrid-material systems, without expert modelling, intervention or other per-robot specialization of the algorithm. Figure 3 demonstrates how our system controlled the diverse robotic platforms towards executing a variety of skills. The system achieved smooth trajectories and succeeded in controlling the pneumatic hand mounted on the UR5 robot to pick up a tool from a glass and use it to push an apple. On the Allegro hand, our system formed a fist. On the HSA platform, it executed a variety of extension and rotation commands. Finally, our method was able to control the low-cost Poppy robot arm to trace the letters MIT. To sum up, across a diverse set of robots, our system could control these systems to perform a variety of long-term skills without any expert modelling or customization.

## Discussion

We have presented a vision-based deep-learning approach that learns to control robots from vision alone, without any assumptions about the robot's materials, actuation or embedded sensors. Across

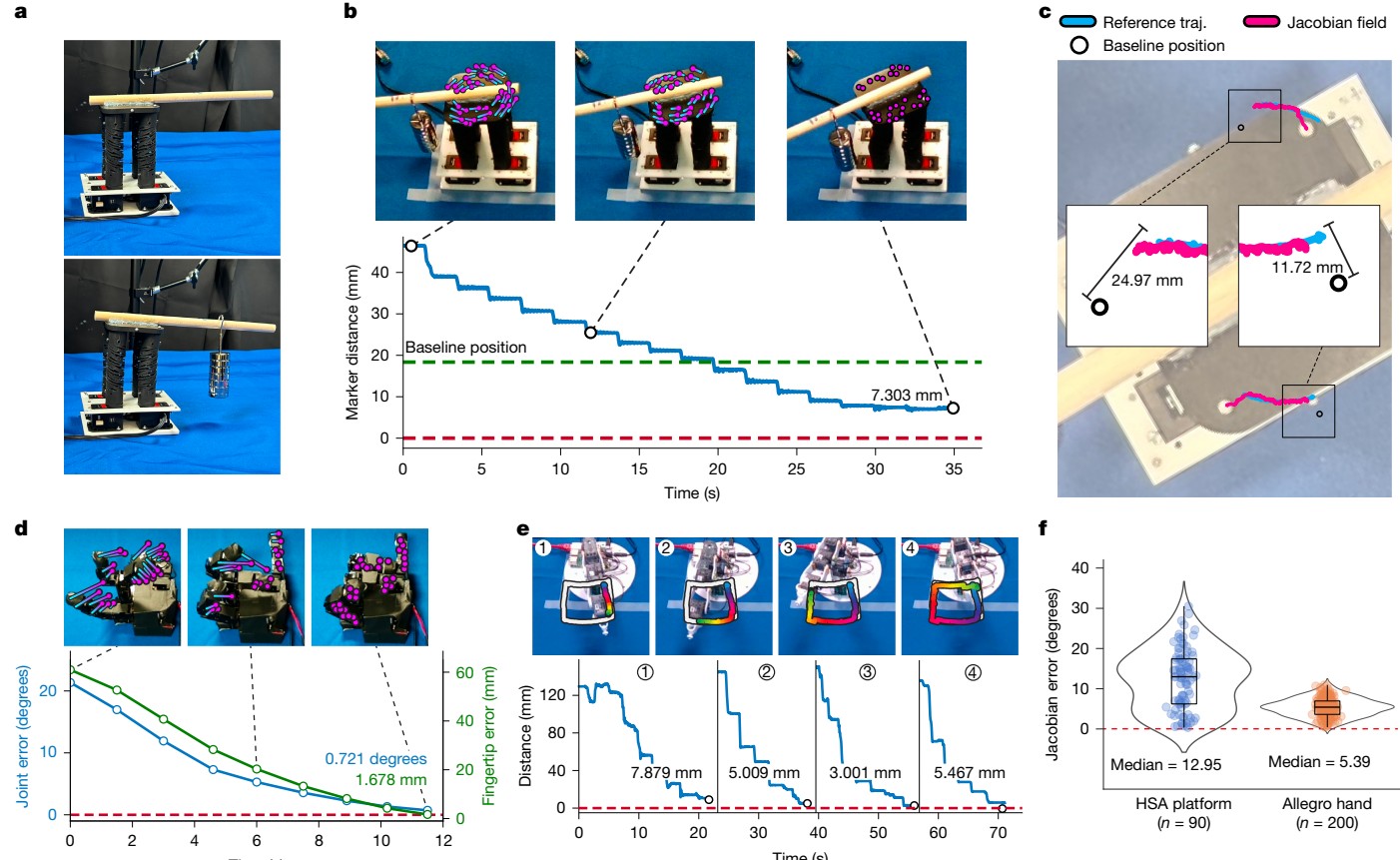

**Fig. 4 | Quantitative analysis and resilience test. a**, We modified the dynamics of the HSA platform. We attached a rod to the platform and appended 350-g weights at a controlled location, causing the platform to tilt in its resting position. **b**, Top, our framework enabled the HSA system with changed dynamics to complete the rotation motion. Bottom, the graph shows the distance from goal over time. **c**, Using a bird's-eye view, we overlaid the completed 3D trajectory (traj.) on top of the starting configuration of the HSA platform. We compared the execution trajectory of our approach with the reference trajectory. This visualization confirmed that our method is able to counteract the physical effects of the weight and stabilize the motion trajectory towards the target path. **d**, The distance from goal of the Allegro hand decreased over time as we executed the motion plan. We measured the distance from goal using both joint errors in degrees and fingertip positions in millimetres. **e**, Top, the reference trajectory is shown in white and the completed trajectory in colours during a square drawing task. Bottom, distance from goal over time using the Poppy robot arm in four trajectory segments. **f**, Comparison of our Jacobian predictions with analytical counterparts computed using physics simulations[47,48]. Our method learnt consistent Jacobian measurements from raw RGB observations.

challenging robotic platforms, ranging from conventional rigid systems to hybrid soft–rigid, 3D-printed, compliant and low-cost educational robots, our framework succeeded in estimating their 3D configuration from vision alone, discovered their kinematic structure without expert intervention, and executed desired motion trajectories with high precision using a single RGB camera. Our system has enabled the modelling and control of 3D-printed, compliant systems without any human modelling and despite substantial changes in their dynamics, replacing a month-long expert modelling process that even so cannot account for changes in material, dynamics or manufacturing tolerances.

Our framework allowed us to control a wide range of robots from vision alone. For this to be feasible, it was crucial that the differential kinematics of the robot could be inferred from vision alone. Some applications of interest may violate this assumption. For instance, when observing mobile legged robots from an external camera, the camera might not observe whether a given leg is touching the ground or not, and thus it cannot determine the motion of the robot as a function of that leg's actuation. Similarly, for dexterous manipulation, sensing contacts with an object is crucial. Conditioning the deep-learning-based inference method for visuomotor Jacobian fields on more sensors, such as tactile ones[42–44], could effectively address this limitation. At test time, only a single camera was necessary to control a robot, but at training time, the use of visuomotor Jacobian fields currently requires multi-view videos.

Our method substantially broadens the design space of robots by decoupling their hardware from their modelling and control. We anticipate that our method will enable the deployment of bio-inspired, hybrid soft–rigid robots, which were previously difficult to model and control. Our method also has the potential to lower the barrier to entry to robotic automation by enabling the control of mass-producible, low-cost robots that lack the precision and sensing capabilities to be controlled using conventional methods. Our method does not currently account for second-order transients and assumes quasi-static motion. This is practical for a wide range of applications[45], including many that involve dexterous hands. Nevertheless, extensions to dynamic effects[46] constitute an exciting area of future work. An in-depth discussion of our modelling assumptions is included in the Supplementary Information.

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

## Methods

### Dataset collection

Our method is fully self-supervised and does not require any manual data annotation. We illustrate the data-collection process in Extended Data Fig. 1a. We captured multiple video streams of the robot executing random actions. Specifically, we set up 12 consumer-grade cameras (Realsense D415 RGB-D, Intel Corporation) that observed the robot from 12 different perspectives.

We obtained intrinsics directly from the cameras. We calibrated camera poses using 3-cm April tags[49]. We denoted the vector of motor set points as $\mathbf{u}$. We first manually selected a safe range for each of the motor channels. To create a single data sample, we randomly selected from a uniform distribution $u$, executed the command and waited for it to settle to a steady state. We then captured images with all 12 cameras, and denoted the time step as $t$.

We then uniformly sampled a change in the motor commands, $\delta\mathbf{u}_t$. The next step command, $\mathbf{u}_{t+1} = \mathbf{u}_t + \delta\mathbf{u}_t$, was then executed on the robot. We again captured images with all 12 cameras and denoted this as time step $t + 1$. This led to a multi-view image dataset, $\{(\mathbf{I}_t^0, \ldots, \mathbf{I}_t^{11})\}_{t=0}^T$, where the superscript denotes the camera index and $t$ denotes the time step. Although our method does not strictly depend on it, leveraging RGB-D cameras that capture depth as well as colour accelerates training owing to further geometry supervision. Finally, we extracted 2D motion information from this dataset using an off-the-shelf optical flow method, RAFT[50], which used as input two consecutive video frames from one of the cameras and computed optical flow $\mathbf{v}_t^i$ between an image captured at times $t$ and $t+1$ using camera $i$, caused by the motor command $\delta\mathbf{u}_t$. Therefore, for each time step $t$, our training dataset is a tuple of the following form:

$$(\{(\mathbf{I}_t^0, \mathbf{d}_t^0, \mathbf{v}_t^0, \mathbf{P}^0, \mathbf{K}^0), \ldots, (\mathbf{I}_t^{11}, \mathbf{d}_t^{11}, \mathbf{v}_t^{11}, \mathbf{P}^{11}, \mathbf{K}^{11})\}, \delta\mathbf{u}_t), \tag{1}$$

with RGB image $\mathbf{I}_t^i$, depth $\mathbf{d}_t^i$, optical flow $\mathbf{v}_t^i$, pose $\mathbf{P}^i$ and intrinsics $\mathbf{K}^i$ for the $i$-th camera, as well as the change in robot command $\delta\mathbf{u}_t$.

Details on how we sampled robot commands during data collection, including discussion of exploration strategies and implications for generalization, are provided in the Supplementary Information.

### Neural 3D reconstruction and neural scene representation

Given a single image, we used deep learning to reconstruct both the proposed Jacobian field and a neural radiance field. Both the Jacobian field and the neural radiance field are functions that map a 3D coordinate $\mathbf{x}$ to the system Jacobian or the radiance and occupancy, respectively.

We followed pixelNeRF[38] to reconstruct both these representations. Given an image $\mathbf{I} \in \mathbb{R}^{H \times W \times 3}$ with height $H$ and width $W$, we first extracted a 2D feature volume $\mathbf{W} \in \mathbb{R}^{H/p \times W/p \times n}$, where $p$ indicates downsampling resulting from convolutions with stride larger than 1. Suppose we want to predict the Jacobian $\mathbf{J}$, radiance $\mathbf{c}$ and density $\sigma$ at a 3D coordinate $\mathbf{x}$. We first project that 3D coordinate onto the image plane using the known camera calibration as $\pi(\mathbf{x})$. We then sample the feature volume at the resulting pixel coordinate using bilinear interpolation $\mathbf{W}(\pi(\mathbf{x}))$. We finally predict the Jacobian $\mathbf{J}$, radiance $\mathbf{c}$ and density $\sigma$ using a fully connected neural network FC:

$$(\mathbf{J}, \mathbf{c}, \sigma) = \text{FC}(\mathbf{W}(\pi(\mathbf{x})), \gamma(\mathbf{x})), \tag{2}$$

where $\gamma(\mathbf{x})$ denotes sine-cosine positional encoding of $\mathbf{x}$ with six exponentially increasing frequencies[33].

Further discussion on the limitations of pixelNeRF in occluded regions, including potential improvements through probabilistic models[51] and global feature conditioning[52], as well as clarifications regarding the coordinate system of our Jacobian field predictions, are provided in the Supplementary Information.

### Training using differentiable rendering

We illustrate the training loop of our system in Extended Data Fig. 1b. In each forward pass, we sampled a random time step $t$ and its corresponding training tuple, as described in equation 1. We then randomly picked 2 of the 12 cameras and designated one as the source camera and one as the target camera. The key idea of our training loop is to predict both the image and the optical flow observed by the target camera given the input view $\mathbf{I}_{\text{input}}$ and the robot action $\delta\mathbf{u}_t$. Both image and optical flow of the target view are generated from the radiance and Jacobian fields by volume rendering[33]. The following discussion closely follows that of pixelNeRF[38].

We first parameterized the rays through each pixel centre as $\mathbf{r}(s) = \mathbf{o} + s\mathbf{e}$, with the camera origin $\mathbf{o} \in \mathbb{R}^3$ and the ray unit direction vector $\mathbf{e} \in \mathbb{R}^3$. We then used volume rendering to predict RGB $\hat{\mathbf{I}}$ and depth $\hat{\mathbf{d}}$ images:

$$\hat{\mathbf{I}}(\mathbf{r}) = \int_{t_n}^{t_f} T(t)\sigma(t)\mathbf{c}(t)\, dt, \tag{3}$$

$$\hat{\mathbf{d}}(\mathbf{r}) = \int_{t_n}^{t_f} T(t)\sigma(t)t\, dt, \tag{4}$$

where $T(t) = \exp(-\int_{t_n}^{t} \sigma(s)\, ds)$ accounts for occlusion through alpha-compositing, that is, points that are closer to the camera with a non-zero density $\sigma$ will occlude those points behind them. For each ray $\mathbf{r}$ of the target camera, we then densely sampled 3D points between near ($t_n$) and far ($t_f$) depth bounds. For each 3D point $r(t) \in \mathbb{R}^3$, we obtained its density $\sigma$ and colour $\mathbf{c}$, and Jacobian $\mathbf{J}$ from equation 2. The notation $\mathbf{I}(\mathbf{r})$ selects the pixel in the image $\mathbf{I}$ that corresponds to the ray $\mathbf{r}$.

Predicted optical flow $\hat{\mathbf{v}}(\mathbf{r})$ is also computed by volume rendering. For every 3D point along a ray, we use the Jacobian quantity to advect the 3D ray sample through $\mathbf{r}(t) + \mathbf{J}(t)\delta\mathbf{u}$. Then we applied alpha compositing to both original 3D ray samples and their advected counterparts to obtain $\hat{\mathbf{x}}(\mathbf{r}), \hat{\mathbf{x}}^+(\mathbf{r}) \in \mathbb{R}^3$:

$$\hat{\mathbf{x}}(\mathbf{r}) = \int_{t_n}^{t_f} T(t)\sigma(t)\mathbf{r}(t)\, dt, \tag{5}$$

$$\hat{\mathbf{x}}^+(\mathbf{r}) = \int_{t_n}^{t_f} T(t)\sigma(t)(\mathbf{r}(t) + \mathbf{J}(t)\delta\mathbf{u})\, dt. \tag{6}$$

Finally, to obtain $\hat{\mathbf{v}}(\mathbf{r})$, we projected $\hat{\mathbf{x}}(\mathbf{r}), \hat{\mathbf{x}}^+(\mathbf{r})$ to the 2D image coordinate using camera intrinsic and extrinsic parameters and computed the positional difference,

$$\hat{\mathbf{v}}(\mathbf{r}) = \hat{\mathbf{x}}^+(\mathbf{r})_{\text{image}} - \hat{\mathbf{x}}(\mathbf{r})_{\text{image}}. \tag{7}$$

**Supervising robot geometry using RGB-D renderings.** To predict the RGB and depth images captured by the target camera, we relied on the radiance field components, colour field $\mathbf{c}$ and density field $\sigma$, in equation 2. The predictions for the RGB image and depth image observed by the target camera are obtained by alpha-compositing the RGB colours and sample depths for each pixel according to equation 4. For each target image with its corresponding pose $\mathbf{P}$, we computed losses

$$\mathcal{L}_{\text{RGB}} = \sum_{\mathbf{r} \in \mathcal{R}} \|\hat{\mathbf{I}}(\mathbf{r}) - \mathbf{I}(\mathbf{r})\|_2^2, \tag{8}$$

$$\mathcal{L}_{\text{depth}} = \sum_{\mathbf{r} \in \mathcal{R}} \|\hat{\mathbf{d}}(\mathbf{r}) - \mathbf{d}(\mathbf{r})\|_2^2, \tag{9}$$

where $\mathcal{R}$ is the set of all rays in the batch. Minimizing these losses trained our model to recover the correct density values and thus the robot

geometry. Note that the depth loss is optional and neural radiance fields are generally trained without it[33,38,51], but because consumer-grade RGB-D cameras are readily available, we relied on this extra signal.

**Supervising the Jacobian field by predicting 2D motion.** We computed a 2D motion loss using ground truth motion tracks to supervise the Jacobian field:

$$\mathcal{L}_{\text{motion}} = \sum_{\mathbf{r} \in \mathcal{R}} \|\hat{\mathbf{v}}(\mathbf{r}) - \mathbf{v}(\mathbf{r})\|_2^2. \tag{10}$$

Minimizing this loss trained our model to predict the correct system Jacobian at each 3D point.

Differentiable rendering is a convenient source of 3D supervision, but it is not the only one. If high-quality depth cameras or motion-capture systems are available, they can also provide 3D information. We discuss alternative supervision strategies for the 3D Jacobian field in the Supplementary Information, including the use of RGB-D data with geometric representations such as occupancy and signed distance fields[53,54].

We provide further analysis on the value of a 3D representation in the Supplementary Information, including how it enables demonstration transfer from unseen viewpoints using shape-based distances[55,56], and how it resolves motion ambiguities inherent to 2D observations[50,57].

## Details of the Jacobian field

Our Jacobian field is a dense, spatial 3D generalization of the conventional system Jacobian in the context of dynamical systems. In this section, we mathematically describe the motivations and insights of our parameterization. We first derive the conventional system Jacobian. Consider a dynamical system with state $\mathbf{q} \in \mathbb{R}^m$, input command $\mathbf{u} \in \mathbb{R}^n$ and dynamics $\mathbf{f} : \mathbb{R}^m \times \mathbb{R}^n \mapsto \mathbb{R}^m$. On reaching a steady state, the state of the next time step $\mathbf{q}^+$ is given by

$$\mathbf{q}^+ = \mathbf{f}(\mathbf{q}, \mathbf{u}). \tag{11}$$

Local linearization of $\mathbf{f}$ around the nominal point $(\bar{\mathbf{q}}, \bar{\mathbf{u}})$ yields

$$\mathbf{q}^+ = \mathbf{f}(\bar{\mathbf{q}}, \bar{\mathbf{u}}) + \frac{\partial \mathbf{f}(\mathbf{q}, \mathbf{u})}{\partial \mathbf{u}} \bigg|_{\bar{\mathbf{q}}, \bar{\mathbf{u}}} \delta \mathbf{u}. \tag{12}$$

Here, $\mathbf{J}(\mathbf{q}, \mathbf{u}) = \partial \mathbf{f}(\mathbf{q}, \mathbf{u})/\partial \mathbf{u}$ is known as the system Jacobian, which is the matrix that relates a change of command $\mathbf{u}$ to the change of state $\mathbf{q}$. Conventionally, modelling a robotic system involves designing a state vector $\mathbf{q}$ that completely defines the robot state and then embedding sensors to measure each of these state variables. For example, the piece-wise-rigid morphology of conventional robotic systems means that the set of all joint angles is a full state description, and these are conventionally measured by an angular sensor in each joint. However, these design decisions are challenging for soft and hybrid soft–rigid systems. First, instead of discrete joints, large parts of the robot might deform. Embedding sensors to measure the continuous state of a deformable system is difficult, both because there is no canonical choice for sensors universally compatible with different robots and because their placement and installation are challenging. Next, designing the state is difficult compared with a piece-wise rigid robot, for which the state vector can be a finite-dimensional concatenation of joint angles. The state of a continuously deformable robot is infinite-dimensional owing to continuous deformations.

Our Jacobian field solves these challenges. First, the combination of Jacobian and neural radiance fields is a complete representation of the robot state; it encodes the position of every 3D point of the robot, as well as its kinematics (how that 3D point would move under any possible action). This relieves us of the need to manually model a robot

state $\mathbf{q}$. Second, we note that, for many robotic systems, it is possible to infer their 3D configuration from vision alone. Even if parts of the robot are occluded, it is often still possible to infer their 3D position from the visible parts of the robot, which is similar to how observing the back of a human arm allows us to infer what the occluded side will look like. In this study, we inferred the state completely from a single camera, but it is straightforward to add more cameras to achieve better coverage of the robot[38].

We now derive the connection of the Jacobian field and the per-camera 2D optical flow we use for its supervision. Rearranging equation 12 yields

$$\mathbf{q}^+ - \mathbf{f}(\bar{\mathbf{q}}, \bar{\mathbf{u}}) = \frac{\partial \mathbf{f}(\mathbf{q}, \mathbf{u})}{\partial \mathbf{u}} \bigg|_{\bar{\mathbf{q}}, \bar{\mathbf{u}}} \delta \mathbf{u}. \tag{13}$$

In practice, our nominal point represents a steady state, because we can wait for the robot system to settle. Then, $\mathbf{f}(\bar{\mathbf{q}}, \bar{\mathbf{u}})$ is approximately $\bar{\mathbf{q}}$. We consolidate $\delta\mathbf{q} = \mathbf{q}^+ - \bar{\mathbf{q}}$ to express the change in robot state $\delta\mathbf{q}$ as a function of the system Jacobian:

$$\delta\mathbf{q} = \frac{\partial \mathbf{f}(\mathbf{q}, \mathbf{u})}{\partial \mathbf{u}} \bigg|_{\bar{\mathbf{q}}, \bar{\mathbf{u}}} \delta \mathbf{u}. \tag{14}$$

We define the dense 3D position of every robot point as the state of the robot. Consequently, the change in robot state $\delta\mathbf{q}$ can be interpreted as the 3D velocity field that moves these points under the action $\delta\mathbf{u}$. The 3D velocity field $\delta\mathbf{q}$ can be measured as 2D pixel motions $\mathbf{v}^i$ across all camera views using off-the-shelf optical-flow and point-tracking methods. Given a training data sample $(\mathbf{v}^i, \delta\mathbf{u}, \mathbf{K}^i, \mathbf{P}^i)$, our visuomotor Jacobian field $\mathbf{J}(\mathbf{x}, \mathbf{I})$ associates the two signals $(\delta\mathbf{q}, \delta\mathbf{u})$ through $\hat{\mathbf{v}}^i = \text{render}\left(\mathbf{J}(\mathbf{X}, \mathbf{I}), \delta\mathbf{u}, \mathbf{K}^i, \mathbf{P}^i\right)$ using equation 7, where $\mathbf{X}$ represents samples of 3D coordinates from the neural field.

To sum up, our Jacobian field leverages visual motion measurements as a learning signal and can be trained self-supervised purely by observing robot motion under random actions with multi-view cameras. It directly relates the change in robot state $\delta\mathbf{q}$, defined as the 3D motion field that advects every 3D point of the robot according to the action $\delta\mathbf{u}$, to the motion in 2D-pixel space observed by multiple cameras. This provides a signal for learning which part of 3D space is sensitive to a particular command of the robotic system, and enables control by specifying the desired motion of any robot point in 2D or 3D.

## Trajectory tracking

Our MPC algorithm supports both 2D and 3D tracking tasks. The main differences lie in the choice of state representation and cost function. In both cases, the goal is to find the control input that best aligns the current observation with the next waypoint. Detailed procedures for demonstration preprocessing, state encoding and command optimization are provided in the Supplementary Information.

**2D Trajectory tracking.** For 2D tracking, we extracted point tracks from demonstration videos using TAPIR[58], guided by segmentation masks from a foundation model[59]. At test time, TAPIR features were used to match or propagate points across frames. Jacobian fields predicted the motion for each pixel, and commands were optimized to minimize L2 distance to the target point locations. We accounted for visibility masks to ignore occluded keypoints.

**3D Trajectory tracking.** For 3D tracking, we lifted RGB video frames to 3D point clouds using PixelNeRF. Each waypoint was stored as a 3D point set sampled from high-density regions of the reconstructed volume. We optimized robot commands by minimizing the Wasserstein-1 distance[55,56] between the current and target point clouds. The Jacobian field was queried at each point to predict the advected cloud under a given control command.

## Evaluations

**Ablative baselines. Jacobian parameterization ablations.** To understand why our Jacobian field enables better generalization and sample efficiency than direct scene flow prediction, we analysed the inductive biases embedded in its formulation. In particular, we studied how linearity[45], spatial locality[60] and compositionality[60,61]—three key properties of mechanical systems—are naturally captured by our parameterization. We complemented these insights with experimental comparisons against a direct flow prediction baseline across real and simulated robotic systems[45,62]. These results demonstrated that our approach generalizes substantially better than the alternatives to unseen robot configurations and commands. Full theoretical observations and experimental results are provided in the Supplementary Information.

**Neural-rendering ablations.** We ablated the neural-rendering part of the Jacobian fields by assuming that the model is given depth as input. Keeping the rest of the architecture unchanged, we trained a pixel-aligned 3D Jacobian fields model. This model was supervised using the same losses on depth and flow. We found that the baseline failed to disentangle the sensitivities of different surface points to different actuators, predicting incorrect Jacobians, probably because of a lack of multi-view supervision. Analysis and results are included in the Supplementary Information.

**Analytical baselines. Allegro hand.** We compared our learnt Jacobian field with an analytical Jacobian derived from the Allegro hand's kinematic model using the Drake simulator[47]. By aligning the predicted and analytical Jacobian fields through Procrustes analysis[63], we computed angular deviations between corresponding command-channel vectors. Our model achieved strong agreement, with an average error of 7° (Extended Data Fig. 6 and Extended Data Table 4). Details of the set-up and evaluation procedure are provided in the Supplementary Information.

**HSA platform.** We benchmarked our model against a 2D analytical-dynamics model of an HSA platform from previous work[48] that does not account for 3D twisting. We aligned the analytical and predicted Jacobian fields in 2D and evaluated vector alignment across sampled points. Our method robustly handled 3D motion and generalized beyond the limitations of the analytical baseline. Full methodology and results are provided in the Supplementary Information.

**Robustness analysis. Robustness against visual occlusions.** To improve the robustness to occlusion and background variation, we applied domain randomization techniques[64–66] during training by overlaying segmented robot foregrounds onto randomized backgrounds and natural scenes[67]. Segmentation masks were obtained using the Segment Anything model[59]. These augmentations improved the resilience of our neural 3D reconstruction module to visual clutter and partial occlusion. Empirical and quantitative results confirmed that depth and Jacobian predictions remained accurate despite such perturbations (Extended Data Fig. 3 and Supplementary Information).

**Robustness against scene perturbations.** We evaluated the ability of our system to generalize to substantial physical scene changes by testing control performance in a modified environment with altered geometry and appearance. Even with large occluders, such as cardboard fences, introduced at test time, our method accurately tracked 3D trajectories using the Allegro hand, achieving a median joint error of 2.89°. More information and experimental set-up are provided in the Supplementary Information.

## Robot systems

We evaluated our method across four diverse robot platforms that had different actuation, morphology and modelling complexity. Full system details, including fabrication, control and sensing, are provided in the Supplementary Information.

**Pneumatic hand.** We used a pneumatically actuated soft robot hand that was 3D-printed in one piece by vision-controlled jetting[6], based on the design in ref. 30. It combined soft PneuNet fingers[68] with a rigid core and was controlled by 15 pneumatic channels. We used two configurations: a stationary hand and a version mounted on a UR5 robot arm.

**Allegro hand.** The Allegro Hand is an anthropomorphic robotic hand with 16 degrees of freedom that is widely used in research[69–72]. It is commercially available and provides a challenging testbed, owing to its high degrees of freedom and mechanical dexterity.

**HSA platform.** This soft robotic platform with four degrees of freedom used handed shearing auxetic actuators[31] that were 3D-printed by digital light projection[73]. The platform's compliant design enables wrist-like motions and linear extension, but accurate modelling is challenging owing to its deformation under external forces[74]. It can be sensorized by fluidic innervation[75] or embedded cameras[76].

**Poppy Ergo Jr.** The Poppy Ergo Jr[32,77] is a low-cost, open-source robot arm with six degrees of freedom built using Dynamixel servos and 3D-printed parts[78]. It is easily affordable but its mechanical tolerances and backlash make it difficult to model accurately[41,79,80].

## Data availability

All the data needed to evaluate the conclusions in the paper are present in the paper or the Extended Data. Our training data are publicly available at https://huggingface.co/datasets/sizhe-lester-li/neural-jacobian-field/tree/main.

## Code availability

The full source code for training the visuomotor Jacobian fields, deploying them on the robots and reproducing the results is available on GitHub at https://github.com/sizhe-li/neural-jacobian-field.

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

**Acknowledgements** We thank H. J. T. Suh for writing suggestions (system dynamics) and T. Chen and P. Agrawal for hardware support (Allegro hand). V.S., S.L.L. and B.C. acknowledge support from the Solomon Buchsbaum Research Fund through MIT's Research Support Committee, the Singapore DSTA under DST00OECI20300823 (New Representations for Vision, 3D Self-Supervised Learning for Label-Efficient Vision), the Amazon Science Hub and the MIT–Google Program for Computing Innovation. S.L.L. was supported by an MIT Presidential Fellowship. A.Z., H.M., C.L. and D.R. acknowledge support from the National Science Foundation EFRI (grant 1830901) and the Gwangju Institute of Science and Technology.

**Author contributions** Contributions to the main ideas and methods were made by S.L.L. (visuomotor Jacobian field and applications to soft robots) and V.S. (visuomotor Jacobian field and applications to soft robots). Contributions to the hardware set-up were made by A.Z. (HSA platform, Poppy robot arm and camera set-up), C.L. (pneumatic hand and FESTO system), B.C. (Poppy robot arm and camera set-up), S.L.L. (Allegro hand and camera set-up) and H.M. (pneumatic hand). Contributions to the formalization of modelling and control challenges in soft robots were made by A.Z., C.L. and D.R. Contributions to the experimental design were made by A.Z. (robustness studies and pneumatic hand), C.L. (pneumatic hand), S.L.L. (all experiments) and V.S. (all experiments). Contributions to the implementation and experimentation of the visuomotor Jacobian field were made by S.L.L. Contributions to data analysis and interpretation were made by S.L.L. (analysis and interpretation), A.Z. (interpretation) and V.S (interpretation). Contributions to computer resources, funding, hardware and facilities were made by V.S. (computer hardware, funding and facilities) and D.R. (robot hardware, funding and facilities). S.L.L., A.Z., B.C., C.L. and V.S. drafted the manuscript; S.L.L., A.Z. and V.S. substantively revised it; and all authors approved the final draft.

**Competing interests** The authors declare no competing interests.

**Additional information**
**Correspondence and requests for materials** should be addressed to Sizhe Lester Li or Vincent Sitzmann.

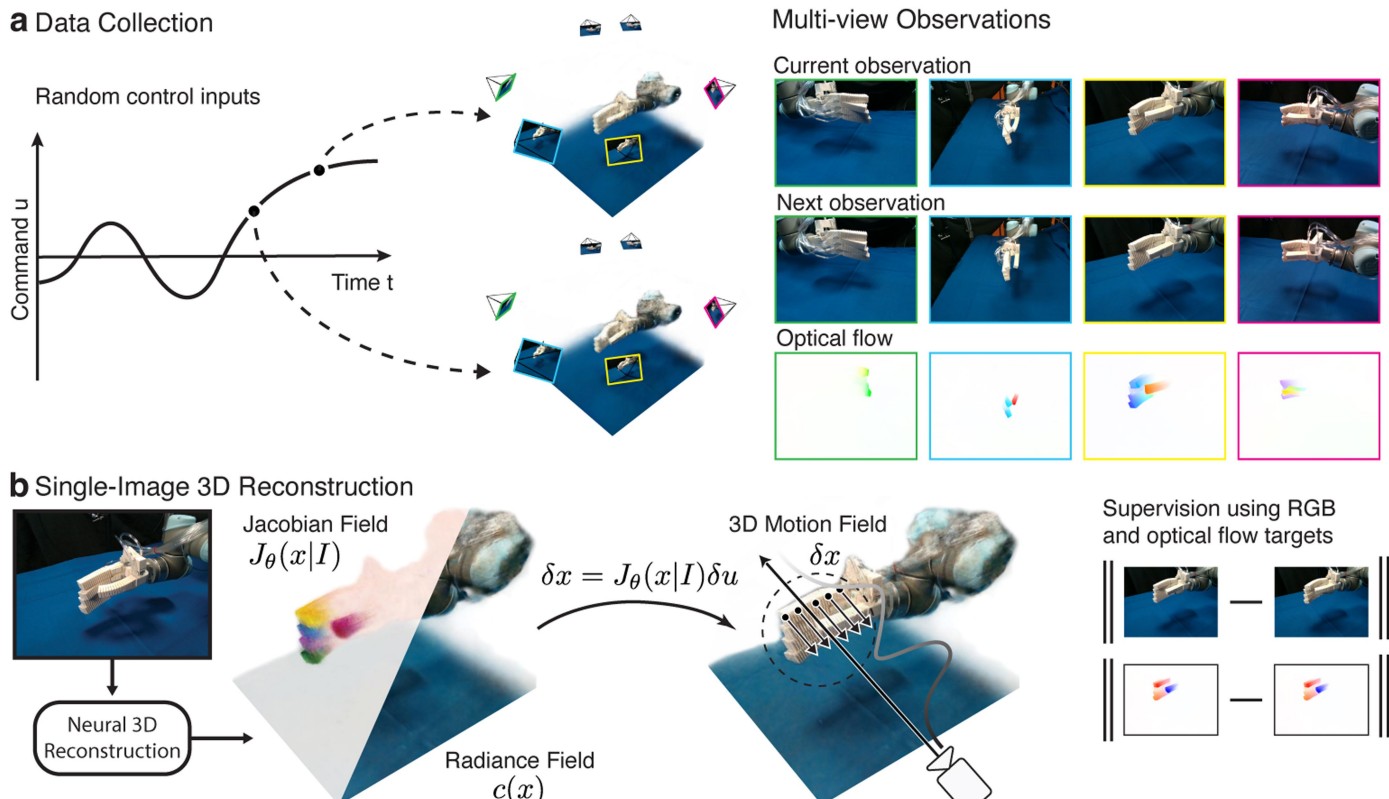

**Extended Data Fig. 1 | Overview of dataset collection, training, and inference processes. a**, Our data collection process samples random control commands to be executed on the robot. Using a setup of 12 RGB-D cameras, we record multi-view captures before each command is executed, and after each command has settled to the steady state. **b**, Our method first conducts neural 3D reconstruction that takes a single RGB image observation as input and outputs the Jacobian field and Radiance field. Given a robot command, we compute the 3D motion field using the Jacobian field. Our framework can be trained with full self-supervision by rendering the motion field into optical flow images and the radiance field into RGB-D images.

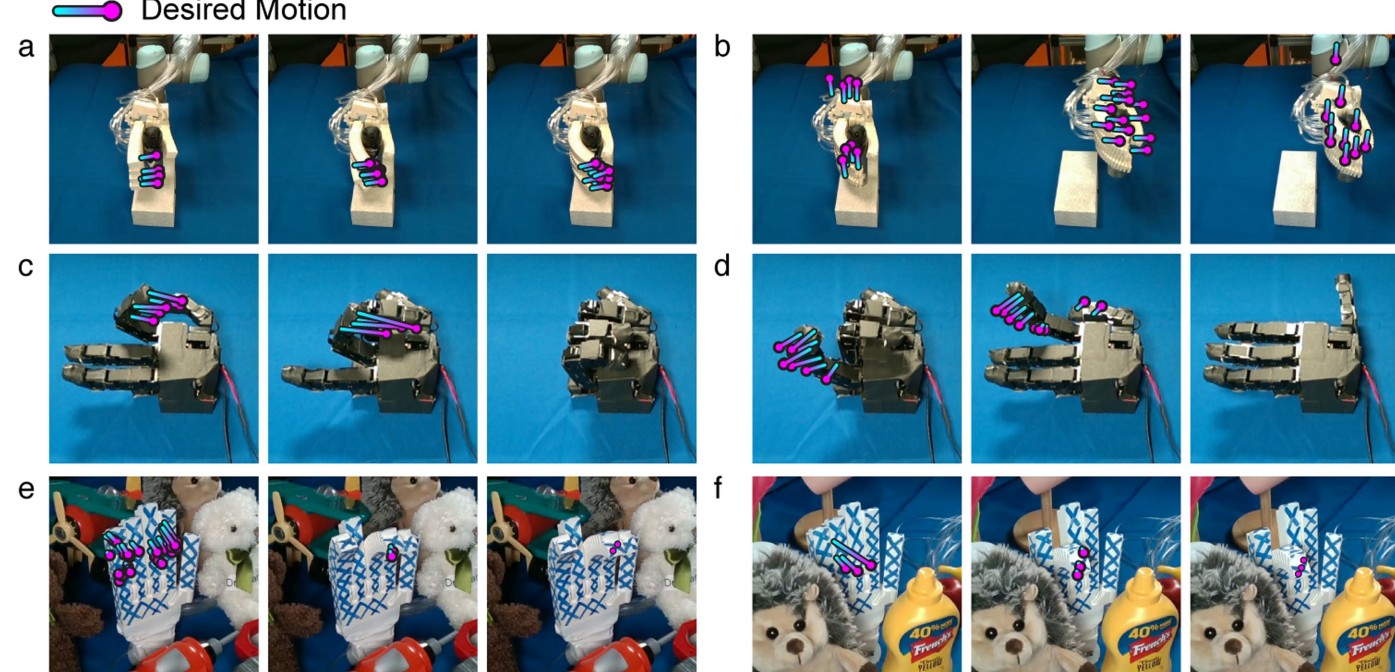

**Extended Data Fig. 2 | Additional evaluation on visuomotor control.**
**(a, b)**, Our framework controls a 3D-printed pneumatic hand to grasp on an object and complete a pick-and-place task. **(c, d)**, Our approach controls the Allegro hand to close and open each finger fully. **(e, f)**, We intentionally perturb the scenes to create occlusion and background changes. We find that the Jacobian field is robust against out-of-distribution scenarios and successfully controls the hand to achieve detailed motions.

|  | RGB Input | Depth Pred. (Input View) | Depth Pred. (Novel View) | Jacobian Pred. (Input View) | Jacobian Pred. (Novel View) |

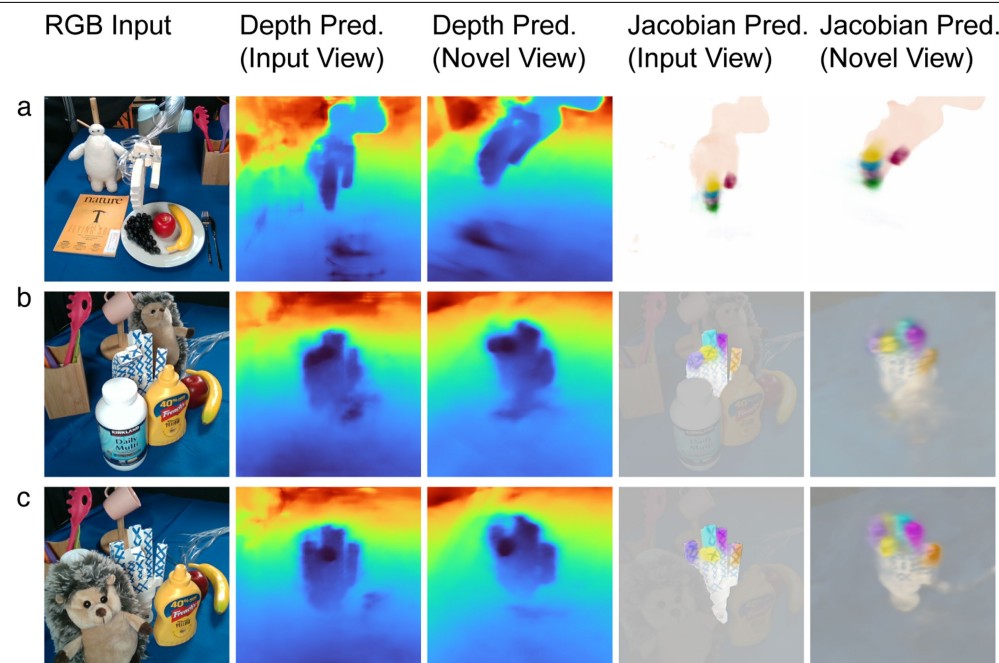

**Extended Data Fig. 3 | Qualitative results on robustness against out-of-distribution scenarios. a**, We perturb the visual scenes by placing objects around the pneumatic hand mounted on the robot arm. Consequently, the input observation is highly out-of-distribution from the training data. We visualize the predictions of depth and Jacobian at both the input and novel viewpoints. We find that our method retains high-quality predictions. **(b, c)**, We place objects around the pneumatic hand to create occlusion. For presentation clarity, we overlay the Jacobian prediction on RGB images to highlight the shapes of the hand with masking.

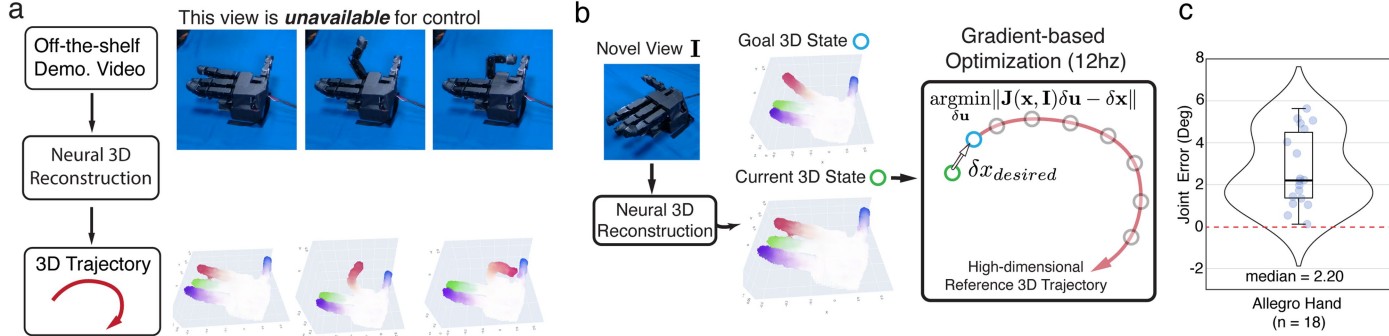

**Extended Data Fig. 4 | 3D enables demonstration transfer between viewpoints. a**, Our model lifts each 2D RGB frame inside the demonstration video to a 3D point cloud (left). The demonstration video comes from a viewpoint that is different from the one available for control (right). **b**, Our 3D representation enables trajectory tracking of demonstration videos from viewpoints unavailable at inference time. We use the Wasserstein-1 distance[56] to measure 3D differences between the current shape and the reference point. **c**, We plot the errors, measured as the joint angle differences between the achieved end state and the final tracking goal.

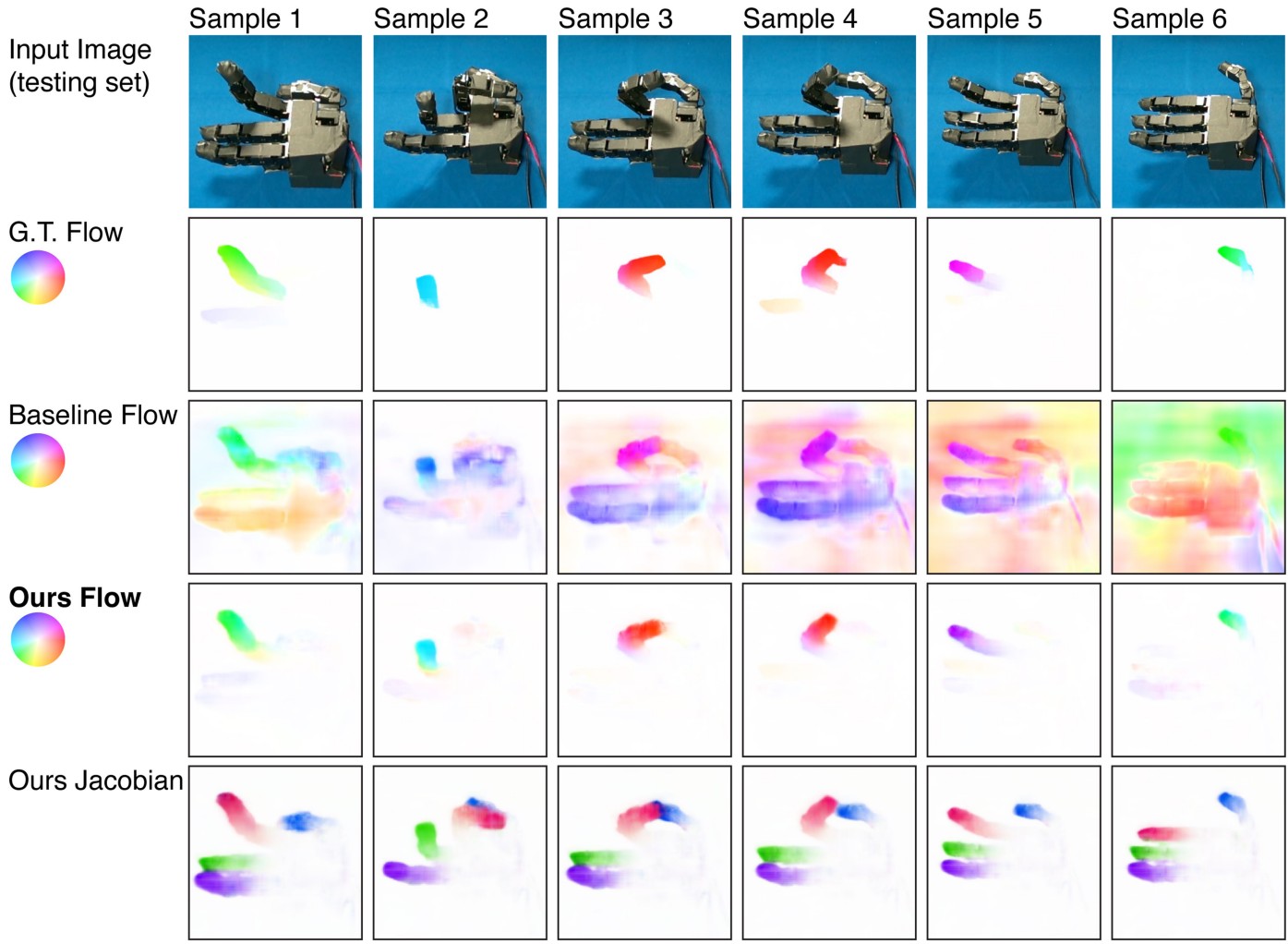

**Extended Data Fig. 5 | Qualitative comparison on test set between our Jacobian model and the direct neural flow baseline.** We evaluate our model and the baseline on the testing samples reported in Extended Data Table 3. Consistent with the numerical results, we qualitatively find that our model can predict correct optical flows on the testing dataset. In comparison, the baseline optical flow model fails to explain out-of-distribution robot commands due to the lack of inductive biases on the locality and symmetry of the dynamical system. In the last row, we visualize the components of our Jacobian model. This validates that the Jacobian model can break down the spatial volume into parts sensitive to each robot finger on the testing dataset. The Jacobian coloration scheme is consistent with Fig. 2.

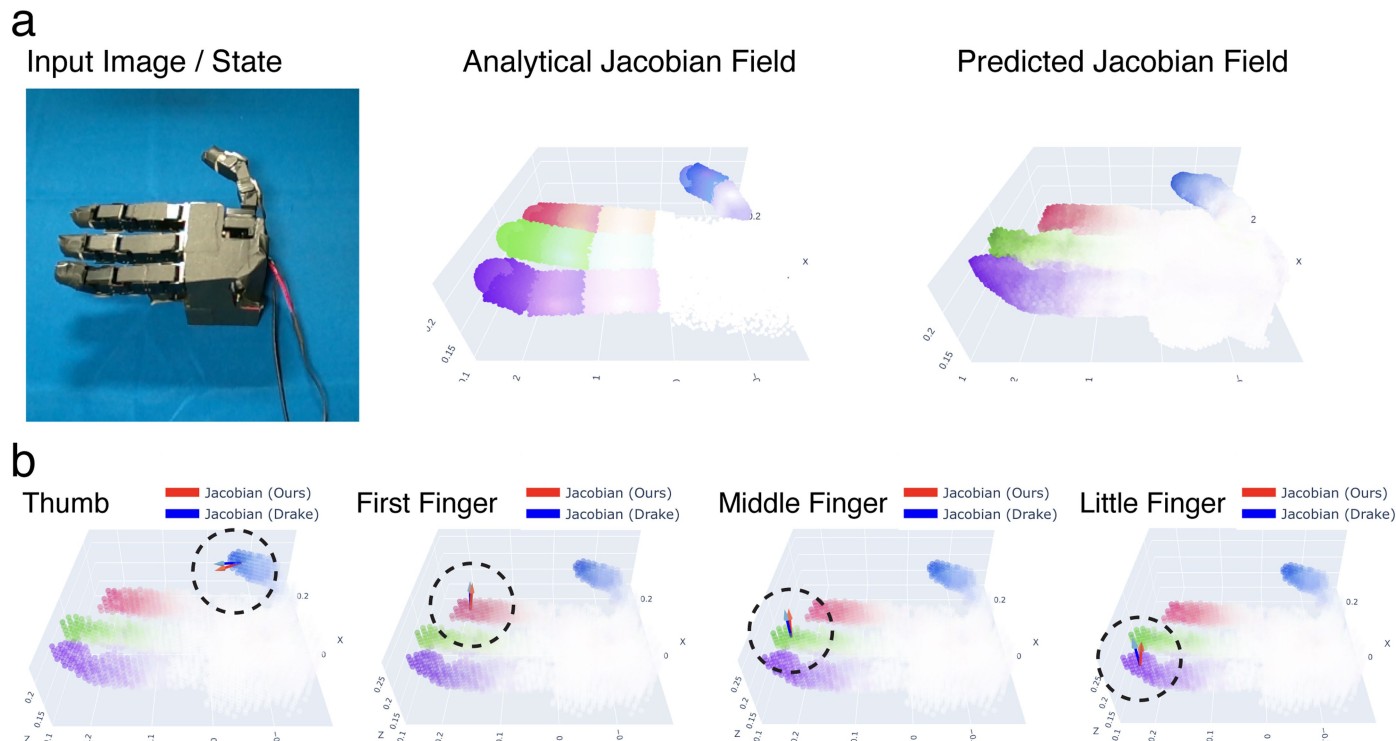

**a**

Input Image / State          Analytical Jacobian Field          Predicted Jacobian Field

**b**

Thumb          First Finger          Middle Finger          Little Finger

**Extended Data Fig. 6 | Visual Jacobian comparison against analytical simulations. a**, We evaluate the quality of our Jacobian predictions against the analytically computed Jacobian field for the Allegro hand in Drake[47]. We find that our model highly matches the analytical Jacobian field, including the low-to-high opacity transition on each finger, which indicates the total change of position upon a command. We highlight that computing the analytical Jacobian relies on the existence of a physics simulator and a URDF file. **b**, We compare each component of the predicted Jacobian (red arrow), queried at a 3D position, with its analytical counterpart (blue arrow). Each column visualizes a different command channel inside the Jacobian, covering all of the robot's motors. Our model achieves high-quality Jacobian predictions, learning only from video observations.

**Extended Data Table 1 | Quantitative evaluation on visuomotor control**

| | a. Allegro Hand | | b. HSA Platform | | | | c. Poppy Arm | d. Pneumatic Hand |
|---|---|---|---|---|---|---|---|---|
| | Angle error (deg) | Location error (mm) | Marker error (mm) | | | | Marker error (mm) | Pressure error (mbar) |
| | | | w/o weight | w/ weight | rotational | translational | | |
| Mean | 2.568 | 3.667 | 5.354 | 7.784 | 7.911 | 3.372 | 5.339 | 5.025 |
| Std | 1.595 | 3.297 | 2.343 | 0.481 | 0.398 | 0.616 | 1.735 | 2.161 |

**a-d**, We measure the distance between the final state achieved by our method and the ground truth desired state. **a**, Joint angle errors in degrees and fingertip positional errors in millimeters. **b**, Motion capture marker location error between the final state achieved by our method and the ground truth recorded in the reference trajectory. **c**, Similar to b, we report errors computed based on the locations of the motion capture markers. **d**, Pressure errors measured in millibars from the 15-channel proportional valve terminal.

**Extended Data Table 2 | Quantitative evaluation on 3D reconstruction**

| | Allegro Hand | | HSA Platform | | Poppy Arm | | Pneumatic Hand | |
|---|---|---|---|---|---|---|---|---|
| | Depth error (mm) | Flow error (pix) | Depth error (mm) | Flow error (pix) | Depth error (mm) | Flow error (pix) | Depth error (mm) | Flow error (pix) |
| Mean | 5.033 | 1.305 | 1.109 | 3.597 | 1.206 | 6.797 | 6.519 | 1.150 |
| Std | 3.080 | 1.504 | 0.576 | 2.779 | 0.619 | 7.287 | 2.028 | 1.520 |

Using testing images from all camera viewpoints, we measure depth prediction errors in millimeters and optical flow prediction errors in pixels on a 640 × 480 image grid. The depth errors are computed as the L2 distance between the ground truth depth value measured by the Intel Realsense D415 RGB-D cameras and the predicted depth value averaged over all pixel locations in the image. The optical flow errors are computed as the L2 distance between the ground truth measured by the point tracker[58] and the prediction averaged over all pixel locations in the image.

**Extended Data Table 3 | Real-world comparison between our Jacobian model and the direct neural flow baseline**

| Optical flow errors (pix) | Pusher Environment | | | Finger Environment |
|---|---|---|---|---|
| | Circle | Moving Left | Moving Up | Closing Second Joint |
| Direct Flow Model | 0.635 | 0.0985 | 0.0969 | 8.651 |
| **Jacobian Model (Ours)** | **0.0247** | **0.00173** | **0.00184** | **0.178** |

We find that our model outperforms the baseline model for the Allegro hand test set. Our model is able to generalize to unseen evaluation samples. We highlight that this baseline model is modeled after the architecture described in[81]. Please see Extended Data Fig. 5 for visual comparisons between the two approaches.

**Extended Data Table 4 | Quantitative comparison against analytical Jacobian (physics simulation)**

|  | Thumb | First Finger | Middle Finger | Little Finger |
|---|---|---|---|---|
| Error Mean (deg) | 10.971 | 8.651 | 5.581 | 6.323 |
| Error Std (deg) | 0.215 | 0.178 | 0.239 | 0.395 |

We evaluate the quality of our model's Jacobian predictions for the Allegro Hand by comparing them to analytical Jacobians computed using the Drake physics simulator[47]. We used five testing samples representing different hand poses and reported the mean and standard deviation of the errors.