## [Peer Review File · Nature]

Controlling Diverse Robots by Inferring Jacobian Fields with Deep Networks

Corresponding Author: Professor Vincent Sitzmann

Version 0:

Reviewer comments:

Referee #1

(Remarks to the Author)

This paper presents an algorithm for inferring Jacobians for the surfaces of arbitrary robotics embodiments with respect to their control signals. The system is highly general, able to represent the dynamics of complicated and non-rigid robots, and the model can be derived via pure self-supervised learning in a multi-camera setup, avoiding the need for the labor-intensive manual modeling that's usually required to obtain Jacobians. The system is demonstrated on several real-world robotics platforms, including non-rigid grippers that are of interest for many real-world applications. Furthermore, the paper demonstrates that the Jacobians can be used to make the robots follow complex trajectories derived from real data, serving as a foundation for general imitation learning.

Regarding originality and methodology, this paper is difficult to review. There are flashes of a truly general algorithm that's well on its way to solve many core robotics problems, yet it's marred by poor writing and a lack of awareness of prior work to properly situate what has been accomplished here.

First of all, the idea of using optical flow to parameterize dynamics models is not new. Perhaps most relevant is Xu et al. 2020, "Learning 3D Dynamic Scene Representations for Robot Manipulation," which also parameterizes a dynamics model using 3D scene flow, and does it not just for the robot, but also for the rest of the scene. Argus et al. 2020 "FlowControl: Optical Flow Based Visual Servoing" shows that optical flow-based dynamics models can be used for more complex control problems than those shown here (i.e. manipulation), and similarly Lee et al. "Aggressive Perception-Aware Navigation using Deep Optical Flow Dynamics and PixelMPC" shows flow-based dynamics models can be used for planning drone movement. None of these papers are cited, and the current paper doesn't contain a clear statement of its intended contributions beyond these. Most of the key claimed properties--such as self-supervised learning of dynamics models--apply to these prior systems as well.

The main novelty in this paper is the use of non-rigid objects where hand-designed models are extremely difficult with forward models, and these were not well evaluated with previous papers. However, in this case, the paper goes too far with its claims. The key conjecture is that dynamics models can be parameterized with Jacobians of 3D motion of surface points and predicted from a single frame. Yet strangely, it emphasizes an issue where this is insufficient: backlash. The model computes Jacobians based on 1) instantaneous appearance of the object in the camera, and 2) 3D position obtained from depth cameras. However, the amount backlash depends on historical actions and forces as well. The paper restricts the setting in a rather non-transparent way, saying in the appendix that "In practice, our nominal point represents a steady state, as one can wait for the robot command to settle."--which is simply not true for any task involving unstable manipulation or where speed matters.

Moreover, the inclusion of 3D position as an input to the dynamics model seems strange, as it means that the dynamics model may break if the entire robot is moved in 3D; no experiments are done where the grippers are on the end of an arm, probably because this will cause problems for the model. The highly-restrictive inputs are likely required, however, in order to maintain invariance to distractors, as the model can use the 3D position as a relatively unambiguous cue for the robot's overall pose. As such, I expect the neural network will likely not generalize beyond the scenes it has been trained on. The paper also broadly ignores the fact that local appearance has problems with domain shift: for example, if the robot is manipulating a previously-unseen object, jacobians will change as the gripper comes into contact with the object, and it's unlikely that the neural net will correctly model this due to the reliance on appearance. Some experiments are done to

address robustness (fig. 4 a,b), but these more argue that incorrect Jacobians are sufficient in this particular instance, rather than showing that the model can estimate correct Jacobians under domain shift.

The writing of the methods section emphasizes the NERF aspect, but the experiments do not make it clear if this is actually necessary or even useful. Xu et al. achieves similar dynamics learning from RGB-D and scene flow. Why should this paper be any different? I.e., why not just use standard scene flow to estimate 3D motion directly, and regress to that? This would remove the reliance on multiple cameras. What's more, most of the evaluations are done by following 2D point trajectories, which raises the question of whether 3D is needed at all. Vecerik et al. 2024 "RoboTAP: Tracking Arbitrary Points for Few-Shot Visual Imitation" (also not cited) achieved 3D control by using 2D visual servoing of dense surface points.

On the other hand, one very novel and, in my opinion, highly under-emphasized aspect of this paper is the use of novel point-tracking methods [41] as a method of parameterizing demonstrations. One of the key problems with Xu et al., Argus et al., and Lee et al. is that local flow provides no means of finding correspondence between the current robot state and any demonstrations, and the papers struggle with this a great deal. The current paper spends just a single paragraph positing a solution to this very fundamental problem, stating "The desired 2D motion is computed as the difference between the target and current 2D locations" without giving any explanation of where the 2D targets come from other than that very recent TAP models are used for this. In some cases, the targets appears to be a final configuration, but in other cases, the model follows an entire trajectory, in which the target locations must change throughout the episode. There is no discussion of this in the supplementary. In practice, this 'demonstration following' for surface points is critical for making Jacobians of surface points a sensible representation, and is critical for making the whole algorithm useful. Almost no attention is given to it.

Interestingly, however, Vecerik et al. provides a more thorough analysis of where to get 2D targets (as well as unpublished concurrent work like Wen et al. "Any-point Trajectory Modeling for Policy Learning"), but Vecerik et al. has a serious weakness: it relies on hand-engineered Jacobians in order to convert target points into motor commands. Ironically, this is precisely the problem that's addressed in the current paper. Despite all the problems with writing, I think that there is a fascinating paper here, one which explores how to learn dynamics models for use with trajectory-following methods based on long-term point tracking. However, such a paper needs to 1) acknowledge prior work on using flow to infer dynamics models, and 2) be a more humble empirical look at the what is required to make these dynamics models work with real point-following tasks. The emphasis on NERF needs to be reduced unless the paper empirically demonstrates that it improves things over prior work like Xu et al. More care needs to be given to establishing how and when the components of this model is required to accurately estimate dynamics, and when the 3D and multi-view setups are required to achieve the generalization.

Referee #2

(Remarks to the Author)

****Summary of the key results****

This work develops a unified approach for modeling and controlling diverse robots. This is particularly useful for low-cost/soft robots where the forward dynamics cannot be analytically modeled, and this work provides a self-supervised learning-based alternative. The key idea is to learn a neural network that, given an input image, predicts a radiance field (mapping each query 3D point to a density and color) as well as a jacobian field (mapping each 3D point to a jacobian matrix indicating the linear motion given some motor command). Importantly, this prediction can be supervised using multi-view images and pixe-lwise 2D flows resulting from the robot motion given an arbitrary motor command.

This approach is validated across impressively diverse robots (e.g. ranging from an Allegro hand to a DIY 3D-printed one). Across all of these, the results clearly demonstrate the ability of the proposed approach to (implicitly) model the system dynamics and allowing real-time control given (dense) desired 2D motion trajectories for a set of points.

****Originality and significance: if not novel, please include reference****

There are two key novel insights in this work. First, while the conventional approach of modeling system dynamics is to analytically map motor commands to motion for pre-defined robot parts, this work generalizes this idea for modeling system dynamics via a (neural) function that can predict the effect of motor commands for any 3D point. Second, this work presents a self-supervised approach for learning such a function — relying on multi-view captures of the robot under random motor commands. This combination of an expressive dynamics model (applicable across different robot morphologies) and a scalable learning approach could potentially be useful across robotics, allowing control of robots even if the system dynamics cannot be easily defined analytically.

That said, the current system does have some limitations which may limit its immediate applicability. For example, the system developed here requires (2D) trajectories for control, but it is not obvious how these can be obtained in application scenarios (e.g. for "conventional" piecewise rigid robots, planning tools exist that can output trajectories given just desired end-effector poses). Similarly, there are open questions regarding the generalizability of the learned dynamics model (e.g. if a model is learned in one lab, what all needs to be done to use it in a different lab setup) — more details on this below. Nevertheless, the work presented here can be a great stepping stone to eventual solutions which resolve these issues in practical deployment.

****Data & methodology: validity of approach, quality of data, quality of presentation****

The approach is validated across four different robots, and qualitative results (images in the paper and video) clearly show the ability of the proposed approach to model the dynamics and allow control across these different systems. I think this is an impressive demonstration of the system, and clearly highlights the main claim — that this is a generalizable approach valid across morphologies.

There are some minor concerns regarding the approach, presentation, and experiments, however, and perhaps these can be addressed via a minor revision:

a) It is unclear what the coordinate system in which a point x is defined. Given an image, are the points x defined in the coordinate space of the camera, or is there an assumed canonical coordinate system? If the latter, does this imply that the learned models would not be easily applicable to slight new setups (e.g. moving the robot arm to a new lab), or would at least require some calibration of identify the new camera position w.r.t. the robot-centered canonical coordinate system?

b) The results are primarily qualitative (albeit with some evaluation for how well some desired trajectories are followed). However, for some systems (e.g. the Allegro hand), one could analytically infer the jacobians and report a measure of how well the learned model captures these. Similarly, for some soft robots, an approximate analytical can be defined and a demonstration that the learned model captures the dynamics better (and allows more accurate control) would be helpful in empirically demonstrating the benefits of the approach.

c) The self-supervised learning setup relies on giving random motor commands and observing its effects. However, if the learned model is to be generally useful, it should observe training states across varying configurations of the robot (e.g. different combinations of open fingers). Are random motor commands really sufficient to yield such a diverse distribution? Or is an exploration strategy required to ensure that the robot configuration space can be adequately covered?

d) The neural network is based on PixelNeRF, which uses pixel-aligned features from the input image to inform prediction for any 3D point. However, a common limitation of such a system is that the information for occluded 3D points is still based on visible ones before them (e.g. the radiance fields learned by pixelNeRF in occluded regions are typically blurry) and this may reduce the accuracy of learned dynamics for hidden 3D points. As an alternative, could a single (image-encoded) global latent-conditioned neural field be a more robust solution?

Conclusions: robustness, validity, reliability

Suggested improvements: experiments, data for possible revision

Some additional text/experiments addressing the 4 concerns raised in the “Data & Methodology” paragraph above would be helpful. More concretely: a) some details about the assumptions in defining the coordinate system and its effect in deployment of the system across setups, b) some empirical evaluation, c) responses about sufficiency of random action sampling and justification of the architecture choice.

References: appropriate credit to previous work?

Perhaps “Neural Jacobian Field” is too broad a term — this models jacobian of 3D point motion w.r.t motion commands, but there are other jacobians one could associate with points in space. In fact, a prior with a similar name already exists (“Neural Jacobian Fields: Learning Intrinsic Mappings of Arbitrary Meshes, Aigerman et. al.”), using jacobians to model mesh manifold. Perhaps some more specific name maybe better that captures to robotics-related nature of the learned fields “Neural Actuation Fields” (not a binding suggestion!)?

Clarity and context: lucidity of abstract/summary, appropriateness of abstract, introduction and conclusions

The paper is generally well written, and the abstract, introduction, and conclusions are appropriate. Perhaps some additional discussion on what steps remain in in deploying the current system for actual manipulation tasks maybe helpful.

(Remarks on code availability)

The provided code has adequate instructions for reproducing the models learned in the paper (although the training data is not released, it is promised to be released upon publication).

Version 1:

Reviewer comments:

Referee #1

(Remarks to the Author)

Overall I believe that the rewrites have done a good job of properly positioning the paper within the literature and more clearly stating what has been accomplished, including its limitations. I'd like to reiterate that I think the paper is overall a very interesting direction and potentially widely useful, allowing for system identification for a wide variety of robot morphologies.

Regarding remaining concerns, I stand by my initial claim (broadly acknowledged by reviewers) that taking a 3D coordinate

as input for the Jacobian is something that helps in the restricted scenarios presented in this paper, but may become be a hindrance for more practical scenarios and limits its generality there.

I apologize for mis-stating that there were no experiments on robot arms. There is one such example, but even here the use of the the arm is limited, and overall it remains problematic for the same reasons. Specifically, from the experiments shown, it appears that the robot arm/wrist has only 2 degrees of freedom, with most of the degrees of freedom in the hand. The result is that if a particular point on the robot's surface is at a particular 3D location, then there is very little remaining ambiguity about its Jacobian. This is still far from more practical situations where a robot hand can move its arm/wrist in a full 6 degrees of freedom. In such scenarios, even if a given point on the robot's surface is at a particular location, it may be at a different *orientation*. This, therefore, puts a much heavier burden on the vision part of the network to estimate the robot's 3D pose, as different 3D poses for the same point may have different Jacobians. This kind of setup is common even in warehouse pick-and-place tasks, where obstacles might mean that the robot needs to reach to the same object from a different orientation. The current paper admits this weakness to some extent, but states it as "Relocating the camera to an arbitrary viewpoint in the scene and generalizing to that view." This feels somewhat misleading, as readers may think that this will work for any application with a static camera, when in fact the algorithm will have the same problem if any of the robots embodiments shown here are put on the end of a long arm with 6 degrees of freedom that is free to move arbitrarily through the environment.

Nevertheless, I appreciate the clarification that the model works in camera coordinates rather than scene coordinates; this means it displays more generalization than I had first thought. Such a setup may be practical in industrial settings where a bespoke robot is built for a particular stage of a production line, and so the pose space of the robot can be better explored, and there are strong correlations between 3D position and Jacobian. It also seems plausible that more data will lead to stronger generalization, i.e., the model will eventually learn good jacobians regardless of 3D location by learning a good appearance model. While I still think the discussion could be improved here, I don't think it's a show-stopper.

My other remaining concern is that, while I appreciate the inclusion of table A4 and I think it strengthens the paper's argument. However, it is also a bit of a strawman. The rebuttal seems most concerned with defending regressing to Jacobians (e.g. "This is in stark contrast to parameterizing f as a neural network that directly predicts scene flow given an image and a robot command since the neural network does not model these symmetries and will thus require orders of magnitude more motion observations to adequately model the system dynamics."), but I'm mostly concerned with clarifying the contribution of the NERF. The advantages of the Jacobian formulation are analytically clear; I was well aware of the math presented in Observation 1. My main point is that we can compute depth/scene flow at training time (possibly with a NERF, or with simple depth+flow) and then train a non-nerf model with similar properties to the one here. A sensible ablation, therefore, is to have this model regresses per-pixel Jacobians directly from pixels. As such, I would appreciate an additional baseline in table A4 which regresses Jacobians using the same losses as for the rest of the framework. I would also appreciate if this is done on the pneumatic hand rather than the Allegro, since for the Allegro the 3D coordinate almost determines the Jacobian, requiring very little appearance information to disambiguate.

Overall, however, I think the paper's main flaws are resolved, and I think the community would find this as an interesting starting point for algorithms to model more of the world's dynamics.

(Remarks on code availability)

I have only briefly reviewed the code; it appears to be a reasonably complete implementation of the algorithm with reasonable documentation, although data is not yet provided. I did not try running it.

Referee #2

(Remarks to the Author)

The main concerns in the original review (copied below for reference) were centered around:

- a) quantitative evaluation of the learned jacobian
- b) need for an exploration strategy
- c) applicability to new setups
- d) need of local conditioning in the model

The author response and the updated manuscript does adequately address these.

In particular, for resolving a), the authors provided additional experiments/visualizations comparing their predictions with the analytically computed ones and these are helpful in understanding the system.

For b), the authors highlighted that a random set of actions were indeed empirically sufficient in this study, which is a strong result and makes a case for easy deployment. The modified text explicitly discusses this.

For c), the authors modified the text to clarify the limitations more explicitly and I believe this discussion is sufficient.

Similarly, for d) (which was a relatively minor concern), the arguments from the authors that the current solution works well empirically are convincing.

Overall, I believe the revised manuscript appropriately addresses all the issues raised, and would be happy to recommend

acceptance.

----- Original Review -----

****Summary of the key results****

This work develops a unified approach for modeling and controlling diverse robots. This is particularly useful for low-cost/soft robots where the forward dynamics cannot be analytically modeled, and this work provides a self-supervised learning-based alternative. The key idea is to learn a neural network that, given an input image, predicts a radiance field (mapping each query 3D point to a density and color) as well as a jacobian field (mapping each 3D point to a jacobian matrix indicating the linear motion given some motor command). Importantly, this prediction can be supervised using multi-view images and pixelwise 2D flows resulting from the robot motion given an arbitrary motor command.

This approach is validated across impressively diverse robots (e.g. ranging from an Allegro hand to a DIY 3D-printed one). Across all of these, the results clearly demonstrate the ability of the proposed approach to (implicitly) model the system dynamics and allowing real-time control given (dense) desired 2D motion trajectories for a set of points.

****Originality and significance: if not novel, please include reference****

There are two key novel insights in this work. First, while the conventional approach of modeling system dynamics is to analytically map motor commands to motion for pre-defined robot parts, this work generalizes this idea for modeling system dynamics via a (neural) function that can predict the effect of motor commands for any 3D point. Second, this work presents a self-supervised approach for learning such a function — relying on multi-view captures of the robot under random motor commands. This combination of an expressive dynamics model (applicable across different robot morphologies) and a scalable learning approach could potentially be useful across robotics, allowing control of robots even if the system dynamics cannot be easily defined analytically.

That said, the current system does have some limitations which may limit its immediate applicability. For example, the system developed here requires (2D) trajectories for control, but it is not obvious how these can be obtained in application scenarios (e.g. for “conventional” piecewise rigid robots, planning tools exist that can output trajectories given just desired end-effector poses). Similarly, there are open questions regarding the generalizability of the learned dynamics model (e.g. if a model is learned in one lab, what all needs to be done to use it in a different lab setup) — more details on this below. Nevertheless, the work presented here can be a great stepping stone to eventual solutions which resolve these issues in practical deployment.

****Data & methodology: validity of approach, quality of data, quality of presentation****

The approach is validated across four different robots, and qualitative results (images in the paper and video) clearly show the ability of the proposed approach to model the dynamics and allow control across these different systems. I think this is an impressive demonstration of the system, and clearly highlights the main claim — that this is a generalizable approach valid across morphologies.

There are some minor concerns regarding the approach, presentation, and experiments, however, and perhaps these can be addressed via a minor revision:

a) It is unclear what the coordinate system in which a point ****x**** is defined. Given an image, are the points ****x**** defined in the coordinate space of the camera, or is there an assumed canonical coordinate system? If the latter, does this imply that the learned models would not be easily applicable to slight new setups (e.g. moving the robot arm to a new lab), or would at least require some calibration of identify the new camera position w.r.t. the robot-centered canonical coordinate system?

b) The results are primarily qualitative (albeit with some evaluation for how well some desired trajectories are followed). However, for some systems (e.g. the Allegro hand), one could analytically infer the jacobians and report a measure of how well the learned model captures these. Similarly, for some soft robots, an approximate analytical can be defined and a demonstration that the learned model captures the dynamics better (and allows more accurate control) would be helpful in empirically demonstrating the benefits of the approach.

c) The self-supervised learning setup relies on giving random motor commands and observing its effects. However, if the learned model is to be generally useful, it should observe training states across varying configurations of the robot (e.g. different combinations of open fingers). Are random motor commands really sufficient to yield such a diverse distribution? Or is an exploration strategy required to ensure that the robot configuration space can be adequately covered?

d) The neural network is based on PixelNeRF, which uses pixel-aligned features from the input image to inform prediction for any 3D point. However, a common limitation of such a system is that the information for occluded 3D points is still based on visible ones before them (e.g. the radiance fields learned by pixelNeRF in occluded regions are typically blurry) and this may reduce the accuracy of learned dynamics for hidden 3D points. As an alternative, could a single (image-encoded) global latent-conditioned neural field be a more robust solution?

****Conclusions: robustness, validity, reliability****

****Suggested improvements: experiments, data for possible revision****

Some additional text/experiments addressing the 4 concerns raised in the “Data & Methodology” paragraph above would be helpful. More concretely: a) some details about the assumptions in defining the coordinate system and its effect in deployment of the system across setups, b) some empirical evaluation, c) responses about sufficiency of random action sampling and justification of the architecture choice.

****References: appropriate credit to previous work? ****

Perhaps “Neural Jacobian Field” is too broad a term — this models jacobian of 3D point motion w.r.t motion commands, but there are other jacobians one could associate with points in space. In fact, a prior with a similar name already exists (“Neural Jacobian Fields: Learning Intrinsic Mappings of Arbitrary Meshes, Aigerman et. al.”), using jacobians to model mesh manifold. Perhaps some more specific name maybe better that captures to robotics-related nature of the learned fields “Neural Actuation Fields” (not a binding suggestion!)?

****Clarity and context: lucidity of abstract/summary, appropriateness of abstract, introduction and conclusions****

The paper is generally well written, and the abstract, introduction, and conclusions are appropriate. Perhaps some additional discussion on what steps remain in deploying the current system for actual manipulation tasks maybe helpful. (Remarks on code availability)

The provided code has adequate instructions for reproducing the models learned in the paper (although the training data is not released, it is promised to be released upon publication).

(Remarks on code availability)

The provided code has adequate instructions and tutorials for reproducing the models learned in the paper (although the training data is not released, it is promised to be released soon).

Author Response: Unifying 3D Representation and Control of Diverse Robots with a Single Camera

1 General Response to Reviewers and Editors

We sincerely thank the reviewers and editor for their time, effort, and insightful suggestions, which have significantly strengthened our work. We are pleased that the reviewers generally appreciated the paper: a novel exploration of learning dynamics models for trajectory-following methods (R1); a method with potential as a foundation for general imitation learning (R1); validated across diverse robotic systems (R2); and a scalable, self-supervised approach with broad applicability across robotics (R2).

Our primary contributions are:

1. *A new class of visual robot dynamics models* that can be autonomously derived from videos and raw motor commands, deployable with a single RGB camera.
2. *A control method* to interface with our dynamics model based on visual point tracking.
3. *An extensive and interdisciplinary study* that brings together new robotic fabrication, modeling, and sensing technologies across a diverse range of robots, including dexterous, soft, and 3D-printed DIY systems.

The multi-disciplinary nature of our work. Recent fabrication advancements promise a new generation of bio-inspired, non-rigid, and low-cost robotics systems. Controlling these systems requires a model of the robot's dynamics and kinematics. However, traditional robot perception, sensing, modeling, and control are incompatible with the wide range of morphologies and actuation mechanisms this new class of robotic systems exhibits. Our framework decouples hardware from the constraints of conventional expert-driven robot modeling and control.

We anticipate that our work will bring together both software and hardware robotics communities to question the design, modeling, and sensing of robots. We anticipate our work will broaden the design space of robotic systems and serve as a starting point for lowering the barrier to robotic automation.

Our response has the following structure:

1. *Overview of new experiments (Sec. 1)* conducted for this revision following reviewers' questions and suggestions
2. *Clarifications* on reviewers' comments (Sec. 2).
3. *Writing revisions* following reviewers' suggestions (Sec. 3).
4. *Details of new experiments (Sec. 4)*.
5. *Point-by-point response to the reviewers (Sec. 5, Sec. 6)*.

The corresponding changes we made in the main manuscript are **highlighted in teal**. We look forward to the feedback and owe many thanks to the reviewers and editors for their insightful suggestions that strengthen our work.

Sincerely, the Authors

1 Additional experiments that **justify method designs through the lens of generalization**.

We agree with the reviewers that additional experiments will justify key design decisions, such as constructing a 3D model and proposing the Jacobian field for modeling dynamics. We agree that additional quantitative results strengthen our paper. In the revised paper, in addition to the many new figures and tables in the Methods sections, we added **a new violin plot in a revised Figure 4** in the main paper, quantitatively comparing our predicted Jacobians against the analytical models.

(R1, R2) Investigation 1: 3D modeling enables demonstration transfer and eliminates motion ambiguity between views.

R1: "Why is 3D modeling needed at all? The evaluations are done by following 2D point trajectories."

1. **Inference-time value of 3D.** We agree with Reviewer 1 that evaluating our system on robotic tasks that uniquely require a 3D representation could justify our design further.
 - (a) We added an experiment that shows our approach enables *demonstration transfer between camera viewpoints*. Given a demonstration video recorded from a no longer available viewpoint, our approach can still control the Allegro Hand from a new viewpoint to follow this demonstration video.
 - (b) This task uniquely requires a *consistent 3D representation*. Our approach lifts the demonstration video and the novel viewpoint observation to a shared 3D state space, a consistent description of the state between viewpoints.
2. **Training-time value of 3D.** We emphasize that the value of 3D can also be seen at training time. 3D modeling eliminates **motion ambiguity between view points**.
 - (a) The motions of robot commands might be indistinguishable from the 2D optical flow of a single viewpoint, as explained in detail later in **Sec. 4.1 of this document**.
 - (b) Our novel view synthesis objective forces PixelNeRF to learn a consistent 3D representation such that the optical flow predicted at an unambiguous viewpoint is correct.
3. We attached the details of Investigation 1 in **Sec. 4.1 of this document**.
4. We have added our results and analysis in a **new Methods Sec. A.7** that we also discuss in the main text of our revised paper.

(R1, R2) Investigation 2: Jacobian parametrization enables generalization to unseen robot configurations and commands.

- R1: "Why Jacobian parameterization instead of direct 3D scene flow prediction? Why neural rendering (e.g., NERF)"
- R2: "Is an exploration strategy required to adequately cover the robot configuration space?"
- R1: "Would including 3D positions as an input break the dynamics model when the entire robot is moved in 3D?"
- R1: "Can our method infer correct Jacobians in unseen environments? For example, when an unseen object comes into contact with the robotic system."

1. We agree with Reviewer 1 that our Jacobian parameterization can be further justified. We conducted a systematic set of experiments in the real world and reproducible simulations to probe our design decisions. **Our results show that the Jacobian flow parameterization is key to generalization to unseen robot configurations and motor commands at test time.**
2. In concert with empirical evidence, we provide new analyses on the origin of our sample efficiency and out-of-distribution generalization:
 - (a) The *linearity* of the Jacobian enables scale-equivariance and reduces the need to learn scaled versions of the same robot command [19].
 - (b) The *locality* and *compositionality* of the Jacobian enables sample-efficient motion prediction for kinematic chains that would otherwise require an exponential increase in training data as a function of the robot's degrees of freedom [16].
 - (c) Predicting motion in 3D instead of 2D image space makes the mapping of control command to action one-to-one, as opposed to many-to-one, leading to better supervision at training time and ultimately better predictions. This is especially true in the regime of small robot commands.
3. We attached the details of Investigation 2 in **Section. 4.2** of this document.
4. We have added our results and analysis in a **new Methods Sec. A.8** that we also discuss in the main text of our revised paper.

(R2) Investigation 3: Robustness against scene perturbations – adding robot appendages, changing camera viewpoints, changing environment geometry.

R2: "Is the method applicable to slightly new setups (e.g., moving the robot arm to a new lab)?"

1. We acknowledge that there are outstanding challenges for our model, such as training on the Allegro in Lab A and generalizing to any Allegro Hand placed in every Lab B. We have discussed and acknowledged these challenges in **Sec. A.9 of our revised paper**.
2. **New camera configuration.** Our *Investigation 1* shows that our system can control the robot from a viewpoint different than the demonstration video by leveraging the 3D representation.
3. **New scene configuration.** We added an experiment where we significantly perturbed the scene by fencing off an Allegro Hand with cardboard. Our system was able to control the robotic hand successfully (**Fig. 4**).
4. We attached the details of Investigation 3 in **Sec. 4.3 of this document**.
5. We have added our results and analysis in **Sec A.9 of our revised paper**.

(R2) Investigation 4: Evaluations of the learned Jacobian in comparison with physics simulators.

R2: "For some systems, one could analytically infer the jacobians and report a measure of how well the learned model captures these."

1. We conducted experiments comparing our learned Jacobian model against analytical models of the Allegro Hand and the HSA Platform.
2. The analytical Jacobian of the Allegro Hand is computed using Drake [25], a state-of-the-art rigid dynamics simulation.
 - (a) Quantitatively, our predictions achieve a high-quality median error of 5.39 degrees over 200 samples, as shown in the violin plot (**Fig. 4 of this document**).
 - (b) Qualitatively, we plot the motion sensitivity of the analytical Jacobian field. We find that our learned prediction highly agrees with the analytical reference (**Fig. 7 of this document**).
3. The analytical Jacobian of the HSA platform is computed using Ref. [23], a state-of-the-art expert-designed deformable solid simulator for the HSA platform.
 - (a) Quantitatively, our predictions achieve a high-quality median error of 12.95 degrees over 90 samples, as shown in the violin plot (**Fig. 4 of this document**).
 - (b) We emphasize that the expert-designed model is not a perfect reflection of the physical reality of the HSA system. It uses simplified 2D geometry and does not model or support 3D twisting motions.
 - (c) To the best of our knowledge, **our model is the only current solution for modeling the dense volumetric dynamics of the HSA platform**, which supports tilt and twist in 3D. The state-of-the-art analytical model [23] only supports 2D symmetrical robot configurations and control of a rigid body frame.
4. The violin plot is added to a **revised Figure 4** in our paper. Details of our results and analysis are included in **Sec. A.10 of our revised paper**. We attached the details of Investigation 4 in **Section. 4.4 of this document**.

2 Clarifications

We thank the reviewers for their attentive reading, and we realize that some points in our manuscript could benefit from additional clarification.

(R1, R2) Clarification 1: Use of 3D coordinates as input to Jacobian Field.

1. We clarify that our 3D representation is reconstructed **in camera coordinates**; the input coordinate \mathbf{x} to our generalizable NeRF is a 3D camera coordinate, not a 3D coordinate in some global reference frame.

- 2. This means that, as long as the relative position of the camera to the robot arm is close to one of the
camera poses seen at training time, the whole system can be moved to a different location, and our model
will continue to make accurate predictions.
- 3. We validate this in **Investigation 3**, where we evaluated our approach with new camera and scene
configurations.
- 4. We added our clarifications on the coordinate system to **Sec. A.2 of the revised paper**.

**(R1) Clarification 2:** How depth information is used in our work.

- 1. We clarify that **depth is not required or used as input to our system at test time**. We use only a single
RGB camera at test time.
- 2. Depth cameras are not required during training. In fact, generalizable radiance fields, including pixelNeRF,
the backbone of our method, are typically trained only using multi-view RGB images [33, 7, 32, 12].
However, depth information reduces the number of camera perspectives needed at training time and
accelerates training by “skipping” empty space; when available, RGB-D imagery thus strictly improves
training.
- 3. In future work, we plan to fine-tune pre-trained RGB-to-3D foundation models [12], which have the
potential to provide further acceleration and robustness.

**(R1) Clarification 3:** “No experiments are done where the grippers are on the end of an
arm, probably because this will cause problems for the model.”

- 1. Our experiments in the original submission already include scenarios in which appendages are mounted
at the end of an arm (**Figure 3 of our paper**). The pneumatic hand is mounted on a UR5 arm, as also
shown in **Figure 8 of this document**.
- 2. Our model is agnostic to the specific robot morphology. It can model the dynamics of any robot manipulator
whose dynamics are visually observable.

(R1) Clarification 4: Differences from Xu et al. 2020, "Learning 3D Dynamic Scene
Representations for Robot Manipulation" [31].

Remarks from Reviewer 1:

1. "Xu et al. achieves similar dynamics learning from RGB-D and scene flow. Why should this paper be any different? I.e., why not just use standard scene flow to estimate 3D motion directly, and regress to that? This would remove the reliance on multiple cameras."
2. "Xu et al. also parameterizes a dynamics model using 3D scene flow, and does it not just for the robot, but also for the rest of the scene"

To contextualize our contributions and justify our design decisions, we compare our Jacobian-based scene
flow prediction method with the neural direct regression approach, a baseline similar to Xu et al. (2020) [31],
which we agree constitutes highly relevant prior work. As detailed in **Investigation 2 (Sec. 4.2)**, our comparative
experiments demonstrate the generalization benefits of our Jacobian formulation. We hope that these additional
results address the reviewer’s concerns regarding our design choice. Next, we clarify and emphasize the key
differences in the problem setting and methodology compared to those of Xu et al.

**Difference 1: Problem Class.** To the best of our knowledge, Xu et al. tackle a different problem. Our work is
a solution to modeling robots with **unknown** morphology, kinematics, and dynamics directly from **raw motor**
**readings and RGB videos**. In contrast, Xu et al. do not study modeling robotic systems. Instead, Xu et al. study
how a specific type of action (planar pushing) from a **known** class of robotic arms with **known morphology,**
**kinematics, and dynamics** will translate to rigid-body motions on rigid **objects**.

Our solution rests on a problem class that is more general and assumes less about the physical world, which is
a key reason why our system is effective on a wide range of robots with unknown morphology, materials, and
actuation mechanisms. To the best of our knowledge, Xu et al.’s system cannot be used directly to solve the

problem class in our work, such as predicting how the HSA platform or robot fingers would move, given raw
motor commands and visual observations.

**Difference 2: Methodology and Capability.** In addition to the problem class, we cannot find a viable way of
applying Xu et al. to solve problems in our paper due to the methodological differences.

- 1. **Representations of geometry, physical dynamics, and robot commands are different.** Xu et al. make
the strong assumption that robot actions are motion vectors specified on single 3D bounding volumes of
objects in the scene. For example, Xu et al.’s model predicts a single SE(3) transformation for each object
in a 3D bounding box.
 - (a) This assumes that objects in the scene move as rigid bodies.
 - (b) As a result, we are not aware of a viable way to apply Xu et al. to model the continuous, dense, and
non-rigid motions of the physical systems in our study, **because these systems cannot be described**
**by rigid body motions with 3D bounding boxes or volumes.** For example, the HSA platform
requires specifications of a dense flow field to achieve precision control, as shown in Figures 3 and 4
in our paper.
 - (c) In contrast, by not injecting system-specific assumptions and by densely representing the continuous
motions of robot surfaces as neural fields, our model can characterize and control a wide range of
robotic systems, including the highly deformable soft HSA platform and the dexterous Allegro hand.
- 2. **Our Jacobian parameterization dramatically improves sample efficiency and OOD generalization.**
As shown in our additional experiments in **Investigation 2 (Sec. 4.2 of this document)**, our Jacobian
parameterization of scene flow leverages locality, linearity and compositionality of robot dynamics,
leading to the generalization of scene flow predictions to out-of-distribution robot configurations and
motor commands. In contrast, the direct scene flow prediction baseline (similar to the design in Xu et al.)
fails to generalize in these cases.
- 3. **The system presented in Xu et al. requires human-in-the-loop supervision.** As stated in Xu et al., the
motion dataset is created by "an in-house human expert." Ground truth object poses that were used as
supervision signals of rigid body motions were annotated by a human in specialized software, as clarified
in Figure 4 of Xu et al. **In contrast, the method presented in our work is completely self-supervised**
**and does not require human interference:** Our model autonomously learns to model robot dynamics
by randomly sampling and executing motion commands on the robot.

3 Writing

**(R1, R2, Editors) Proposed Change 1: Stronger connections with 2D trajectory tracking.**

- 1. We deeply thank the reviewers and editors for their constructive suggestions. We improved the focus of
our discussion on using novel point-tracking methods for parameterizing demonstrations.
 - (a) We rewrote a large portion of our method description (**Section 2 in our revised paper**) to connect
with the general imitation learning applications that our learned dynamics model can offer (**L99-109,**
**L165-L183**). Following Reviewer 1’s constructive suggestion, we emphasized the following in our
revised texts.
 - i. Our inverse dynamics controller parameterized desired motions densely in 2D image space or 3D
to find robot commands.
 - ii. We find that parameterizing demonstration trajectories as dense point motions is the key to
controlling a diverse range of robotic systems, as motions of deformable and dexterous robots
cannot be well-constrained via rigid transformations specified on a single 3D frame.
 - iii. Our parameterization enables a wide range of robots to imitate video-based demonstrations.
 - (b) Our new related work section in **a new Methods Sec A.12** clarifies our connections with trajectory
tracking and differential kinematics methods.
- 2. As reviewers remark, our system only solves the *system identification and control* problem, not the *motion*
*planning* problem, i.e., it requires 2D/3D trajectories that specify the intended motion at test time. While
we leave a thorough investigation of a scene-flow-based motion planner to future work, as reviewers

remark, elegant work by Vecerik et al. and Wen et al. [28, 30] already show the first steps towards this
goal.
3. While a generalist flow-based motion planner is beyond the scope of the present submission, we are
confident that our model will inspire the community to design motion planners that can integrate with
our proposed Jacobian representation of the kinematics of a robotic system.

**(R1, Editors) Proposed Change 2: Adding a related work section.**

We added a related work section that discusses our similarities and differences with respect to the suggested
references in a **new Methods section A.12**, in particular discussing Xu et al., Wen et al., and Vecerik et al [28,
31, 30].

**(R1) Proposed Change 3: Adding clarifications on our modeling assumptions.**

**In a new Methods section A.6**, we added a detailed discussion on the assumptions made in our framework
about the dynamics of a physical system, including clarifications on backlash issues and the modeling of
second-order transients.

**(R1, Editors) Proposed Change 4: Reduce emphasis on differentiable rendering, clarify the necessity of 3D.**

- 1. **In the new Methods Section A.4** of our revised paper, we discuss alternative ways to supervise our
proposed Jacobian fields. This discussion is placed right after our discussion on the supervision signals of
our current model.
- 2. **In the revised paper, line 85**, we removed "differentiable rendering" from our statement about the
techniques by which our method is uniquely enabled.
- 3. R1 remarks that the value of the differentiable rendering step is unclear. We agree that the emphasis on
differentiable rendering should be reduced: It is indeed true that the critical aspect of our method is the
fact that the Jacobian Field is a *3D Jacobian Field*, not how we obtain 3D (via differentiable rendering or
via RGB-D cameras). We hence followed the reviewer's suggestion and adjusted the paper accordingly to
reduce the emphasis on NeRF. This is reflected in a **new Methods Section A.4** of our revised manuscript.
We clarified why 3D features and 3D Jacobians, not necessarily obtained via neural rendering, are essential
to the proposed approach. We also discuss alternative methods to supervise the Jacobians.
- 4. We clarify that while not *necessary*, differentiable rendering is nevertheless *useful*, as it has several
advantages over an approach that leverages RGB-D cameras to build the 3D Jacobian Field. First, neural
rendering allows us to train our method only from RGB images; **RGB-D cameras are optional**. Second,
differentiable rendering has demonstrated outstanding performance in high-quality 3D reconstruction,
achieving better results than consumer RGB-D cameras [29, 15], though still lagging behind LIDAR and
other high-end capture systems.
- 5. We highlight that our system **does not require RGB-D cameras**. At training time, multi-view RGB
cameras suffice, though if RGB-D is available, it accelerates training and reduces the number of necessary
cameras. At inference time, only a single RGB camera is required.
- 6. We highlight that while neural rendering vs. RGB-D cameras is mostly a practical consideration, the 3D
Jacobian Field parameterization of scene flow is essential to achieving sample-efficient robot control, as
discussed in **Investigation 1 (Sec. 4.1 of this document)**.

**(R1) Proposed Change 5: Adding clarifications and details of control and planning procedures.**

We added details **in a new Methods Section A.6** on our control and planning procedures for following
surface points in video demonstrations. We added pseudocode in the new section. We publicly released our
implementation under <https://github.com/sizhe-li/neural-jacobian-field>.

(R2) Proposed Change 6: Adding clarifications on the details of action sampling strategies.

We provided a discussion called "Details on Command Sampling" in a revised **Methods section A.1** on the details of the sampling strategies used for exploring the distributions of robot configurations during data collection.

(R2) Proposed Change 7: Add discussion of the value of pixelNeRF and its limitations, comparison with other architectural choices for image-to-3D modeling.

Thank you for your insight on PixelNeRF and occlusion! **At the end of Section A.2** of our revised paper, we added a detailed discussion on the limitations of pixelNeRF in the face of occlusion and other architectural choices, such as the global latent-conditioned neural field.

(R2) Proposed Change 8: New method name that clarifies the context of Jacobian.

Thank you for your suggestions on our method name (initially "Neural Jacobian Fields"). We propose "Visuomotor Jacobian Fields." We have this proposal in our revised paper.

4 Details of new experiments that justify method designs through the lens of generalization.

We are excited that both reviewers are interested in how far our system can generalize to unseen inputs. At the same time, both reviewers have provided insightful questions on method designs. **We find that questions about generalization and method design are two sides of the same coin.** Our first three investigations below will answer these questions:

Questions about key design decisions of the method:

- **Q1.** "Why is 3D modeling needed at all? The evaluations are done by following 2D point trajectories." (from Reviewer 1)
- **Q2.** "Why Jacobian parameterization instead of direct 3D scene flow prediction? Why neural rendering (e.g., NERF)" (from Reviewer 1)

Questions about generalization:

- **Q3.** "Is an exploration strategy required to adequately cover the robot configuration space?" (from Reviewer 2)
- **Q4.** "Would including 3D positions as an input break the dynamics model when the entire robot is moved in 3D?" (from Reviewer 1)
- **Q5.** "Can our method infer correct Jacobians in unseen environments? For example, when an unseen object comes into contact with the robotic system." (from Reviewer 1)
- **Q6.** "Is the method applicable to slightly new setups (e.g., moving the robot arm to a new lab)?" (from Reviewer 2)

4.1 Investigation 1: 3D modeling enables demonstration transfer and eliminates motion ambiguity between views. (Q1)

R1: "Why is 3D modeling needed at all? The evaluations are done by following 2D point trajectories."

3D enables demonstration transfer between viewpoints. Our 3D representation enables trajectory tracking of demonstration videos from viewpoints unavailable at inference-time (**Fig. 1**). Given a demonstration video recorded from an unavailable viewpoint, our PixelNeRF lifts each frame to 3D point clouds to form a high-dimensional trajectory (**Fig. 1a**). Using a shape-based distance in 3D [11, 10], our model enables demonstration transfer from unavailable viewpoints. We did not cherry-pick designs for the model predictive controller and used the same control algorithm regardless of whether the reference trajectory is 2D or 3D.

Figure 1: **3D enables demonstration transfer between viewpoints.** **a**, Our model lifts each 2D RGB frame inside the demonstration video to a 3D point cloud (left). The demonstration video comes from a viewpoint that is different from the one available for control (right). **b**, Our 3D representation enables trajectory tracking of demonstration videos from viewpoints unavailable at inference-time. We use the Wasserstein-1 distance [10] to measure 3D differences between the current shape and the reference point. **c**, We quantitatively plot the errors, measured as the joint angle differences between the achieved end state and the final tracking goal.

**Experimental Results.** We test demonstration transfer on the Allegro Hand, whose analytical model is reliable
 for performance analysis. Our framework enables the Allegro hand to track demonstration videos specified in
 unavailable viewpoints. For every demonstration trajectory, we run our method ten times to compute the final
 error’s statistical average and standard deviation, measured as the difference between the final achieved joint
 angles and the ground truth joint angles in degree. We find that our method achieves a high-quality median error
 of 2.2 degrees, as reported in Fig. 1c.

**3D eliminates motion ambiguity between viewpoints** A key challenge of learning robot control from 2D
 motion alone is that 2D observations are inherently *ambiguous* in the sense that many potential 3D motions map
 to identical or almost identical 2D motion. This form of motion ambiguity is a well-studied problem in computer
 vision [26, 4]. For instance, observing the HSA platform directly from the top will lead to vanishing optical flows
 for extending and shortening of the platform. For multi-fingered robots, bending or moving diagonally induces
 almost identical optical flows when observing the robot hand from the side.

Our novel view synthesis objective forces our network to learn a single 3D Jacobian field that correctly predicts
 motion in *all* available camera perspectives, resolving the ambiguity.

4.2 Investigation 2: Jacobian parametrization enables generalization to unseen robot configurations and commands. (Q2, Q3, Q4, Q5)

- **R1:** "Why Jacobian parameterization instead of direct 3D scene flow prediction? Why neural rendering (e.g., NERF)"
- **R2:** "Is an exploration strategy required to adequately cover the robot configuration space?"
- **R1:** "Would including 3D positions as an input break the dynamics model when the entire robot is moved in 3D?"
- **R1:** "Can our method infer correct Jacobians in unseen environments? For example, when an unseen object comes into contact with the robotic system."

Next, we show analysis and experimental results on how the Jacobian parametrization of scene flow enables
 out-of-distribution generalization to unseen motion at test time, as well as improved sample efficiency. We first
 observe that mechanical systems made from continuum solids display *linearity* and *locality* [3, 13, 19, 16],
 providing formal justification for the empirically demonstrated generalization of our system.

**Observation 1: Linearity from local theory of smoothing [19]** We model *differential* kinematics by
 linearizing the system dynamics, which represents 3D motion fields induced by *small* control commands δu . In
 this regime, it is well-known that the 3D motion δx of robot 3D points can be described by the space Jacobian *and*
 *is thus a linear function of control commands*, i.e. $\alpha \delta x = \frac{\partial f}{\partial u}(\alpha \delta u)$. This is powerful, as completely specifying the
 system dynamics for a particular configuration u' requires only $n \times 3$ linearly independent observations of pairs of
 control commands and induced scene flow, as this fully constrains the space Jacobian for a given configuration for
 a system with n control channels. For instance, one need not observe the motions for *both* $-\delta u$ and δu ; it suffices

to observe *one* of them. Similarly, one need not observe δu and a scalar multiple $\alpha\delta u$; again, one of them in the
 training set suffices. This is in stark contrast to parameterizing f as a neural network that directly predicts scene
 flow given an image and a robot command since the neural network does *not* model these symmetries and will
 thus require orders of magnitude more motion observations to adequately model the system dynamics.

**Observation 2: Spatial locality of mechanical systems.** Robot commands often result in highly localized
 spatial motion. Here is an intuitive example:

- • **Locality of independent kinematic chains [16].** Consider a multi-fingered robot hand (Fig. 4),
 commanding the thumb motors in this configuration leaves the little finger still. The manipulator
 Jacobian field as a physical quantity is locally smooth across space.

**Observation 3: Spatial compositionality of mechanical systems.** Robot commands often result in spatial
 motions composed by the influences of individual command channels. Here is an intuitive example:

- • **Compositional kinematic joints [16, 19].** Consider the simple two-joint robot finger in Fig. 4, for a
 material point inside the second segment (as a solid domain [3]), its motion is computed as the integral
 over individual command channels in the Jacobian tensor field. In the example of Fig. 4, the 2D motion
 is the summation over two Jacobian channels.

**Experimental evidence of benefits of locality, linearity, and compositionality.** We now provide
 experimental evidence across a diverse set of real-world and simulated domains to show that the inductive bias of
 our Jacobian parameterization leads to better predictions and out-of-distribution performance.

**Real-world experiments with Allegro Hand.** We benchmark our Jacobian field parameterization with a
 baseline where we directly predict the scene flow δx conditioned on a robot command δu . Specifically, we provide
 an MLP with the per-pixel feature vector as well as the motor command and train it to output the 3D scene flow on
 the same training data and with the same CNN backbone as our Jacobian Field. We compare this baseline against
 our approach both quantitatively and qualitatively for the flow prediction task, reported in Table 1 and Figure 2.
 **This baseline closely resembles the scene flow prediction model in Xu et al. (2020) [31], suggested by**
 **Reviewer 1.**

Optical flow error (pix.)	Sample 1	Sample 2	Sample 3	Sample 4	Sample 5	Sample 6	Mean	Std
Direct Flow Model	12.191	0.466	6.629	10.584	15.808	21.095	11.129	7.164
Jacobian Model (Ours)	0.809	0.365	0.183	0.166	0.370	0.193	0.348	0.244

Table 1: **Real-world comparison between our Jacobian model and the direct neural flow baseline.** We find that our model outperforms the baseline model on the Allegro Hand testing set. Our model is able to generalize to unseen evaluation samples. We highlight that this baseline model is modeled after the architecture described by Xu et al., 2020. Our evaluation samples are named consistently between this table and Figure 2. For detailed visual comparisons between the two approaches, we refer the readers to Figure 2.

The highly dexterous Allegro hand has 16 degrees of freedom, with independent kinematic chains (i.e., four
 fingers). The dexterity of the system creates combinatorial complexity for modeling the configurations of the
 robot hand. In Figure 2, we show that the baseline model fails to predict the correct motions for the robot hand
 on the validation dataset, where our method succeeds. We note that the baseline method fits the training set well,
 suggesting that the problem is generalization. Quantitatively, as shown in Table 1, our model attains significantly
 lower flow prediction error.

**Simulated experiments.** We created a physics-based simulation of two conventional robotic systems [16, 19]
 in 2D. We create our model and the baseline model in the same fashion as our real-world comparison above. Our
 model predicts a 2D Jacobian at a pixel conditioned on an image observation, whereas the baseline model directly
 predicts the 2D flow at a pixel conditioned on the image observation and the robot command. We have added our
 results and analysis in **the new Methods Sec A.8 of our revised paper.**

- 1. **(Pusher Environment [8]; Figure 3)** The environment contains a spherical robotic pusher [8]. The
 robot can move freely in 2D space and is steered by a 2D velocity command $\delta u \triangleq (x, y)$, where $x, y \in \mathbb{R}$.
- 2. **(Dexterous Finger Environment [19]; Figure 4)** The environment contains a 2 degrees-of-freedom
 robot finger. The robot finger is commanded by a 2D joint velocity command $\delta u \triangleq (u_1, u_2)$, where
 $u_1, u_2 \in \mathbb{R}$ control the rotations of each motor respectively.

Figure 2: **Qualitative comparison on test dataset between our Jacobian model and the direct neural flow baseline.** We evaluate our model and the baseline on the testing samples reported in Table 1. Consistent with the numerical results, we qualitatively find that our model can predict correct optical flows on the testing dataset. In comparison, the baseline optical flow model fails to explain out-of-distribution robot commands due to the lack of inductive biases to the locality and symmetry of the dynamical system. In the last row, we visualize the components of our Jacobian model. This validates that the Jacobian model can break down the spatial volume into parts sensitive to each robot finger on the testing dataset. The Jacobian coloration scheme is consistent with Figure 2 in our main manuscript.

**(Results of Pusher Experiment; Fig. 3; Tab. 2)** Training on just two trajectories of the robot moving down
 and moving right. We investigate whether the learned dynamics model generalizes to unseen spatial locations
 and unseen robot command magnitudes and directions.

We find that the Jacobian model can generalize to the whole space of \mathbb{R}^2 configurations, and the whole space
 of \mathbb{R}^2 motion magnitudes and directions. This is substantiated by Fig. 3 and Tab. 2, where we task the robot to
 draw curves and move in directions at locations unseen during training. In comparison, the direct flow prediction
 model fails to generalize to unseen motion magnitudes and directions, and thus is unable to control the robot.

	Pusher Environment			Finger Environment
Optical flow errors (pix.)	Circle	Moving Left	Moving Up	Closing Second Joint
Direct Flow Model	0.635	0.0985	0.0969	8.651
Jacobian Model (Ours)	0.0247	0.00173	0.00184	0.178

Table 2: **Simulated comparison between our Jacobian model and the direct neural flow baseline.** The mean errors are computed by evaluating each trajectory 10 times. We find that our model outperforms the baseline model. Our model is able to generalize to unseen evaluation samples. We highlight that this baseline model is modeled after the architecture described by Xu et al., 2020 [31].

**(Results of Finger Experiment; Fig. 4; Tab. 2)** For training, we create two trajectories that represent just
 two types of commands: rotating the first motor only, rotating the second motor only. For testing, we turn the
 first motor to an unseen configuration and rotate the second motor and vice versa. We ensured that the baseline
 model was not underfitting. As observed in Fig. 4, the baseline model can perfectly reconstruct training samples.

We find that our model generalizes to the out-of-distribution test scene, solving both the forward and the
 inverse problems. In comparison, we found that the baseline model fails at unseen configurations, incorrectly

Figure 3: **Evaluations of Jacobian properties using 2D Pusher Environment.** **a**, Training on just two trajectories of the robot moving down and moving right, both our model and the direct neural flow baseline are able to fit the dataset well. **b**, Our Jacobian model is able to generalize to unseen spatial locations and unseen robot commands. Our model predicts correct optical flow values, as shown by the time overlay. In comparison, the baseline model fails to generalize due to the lack of inductive biases.

predicting motions not only on surfaces of the second finger segment but the first one as well, i.e. not successfully
 disentangling parts of the kinematic chain.

Experimental evidence above, from both the real world and the simulated domain, substantiates our claim –
 the Jacobian parameterization, through capturing spatial locality and spatial symmetry of the robotic system,
 dramatically improves out-of-distribution motion prediction.

4.3 Investigation 3: Robustness against scene perturbations – adding robot appendages, changing camera viewpoints, changing environment geometry. (Q6)
R2: "Is the method applicable to slightly new setups (e.g., moving the robot arm to a new lab)?"

 **Current limitations.** We first humbly acknowledge that the following settings remain challenging, which is
 the holy grail for many research efforts across robotics, computer vision, and machine learning. **Our learned**
 **dynamics model cannot yet handle the following settings.**

- 1. Training on Allegro Hand in Lab A, generalizing to any Allegro Hand placed in every Lab B.
- 2. Relocating the camera to an arbitrary viewpoint in the scene and generalizing to that view.
- 3. Changing the robot’s morphology after the model is trained. E.g., Uninstalling a finger from the Allegro
 Hand; Installing the Allegro Hand to an arbitrary robot arm.

**We have humbly acknowledged these challenges in Sec. A.9 of the revised paper.**

Figure 4: **Evaluations of Jacobian properties using 2D Finger Environment.** **a**, Training on just two trajectories of the robot rotating the first and the second motors, both our model and the direct neural flow baseline can fit the dataset well. **b**, Our Jacobian model can generalize to unseen finger configurations and robot commands. Our model predicts correct optical flow values. In comparison, the baseline model fails to generalize due to the lack of inductive biases.

**Our model makes system identification scalable.** Our system identification is amortized by neural networks
 (PixelNeRF) that encode the features of geometry, kinematics, and dynamics for surface points on the robot. This
 architecture of single image to 3D has since been scaled to large datasets and large models [12, 24], advances that
 our model will benefit from as well. Hence, the identification and perception of the physical systems is scalable in
 our approach. In our original manuscript (**Figure 3, d**), the translational equivariance of convolutional neural
 networks enabled our model to control the pneumatic robot hand when the arm position is out of distribution,
 shown in **Figure 8**.

**Our model deployment does not need camera calibration.** Our model does not require camera calibration
 after training. Our 3D model is constructed in camera coordinates, not world coordinates. The 3D model is trained
 to predict novel view information relative to the current viewpoint without knowing globally the orientation and
 location of the current viewpoint.

**Experimental results on heavy appearance perturbation of the scene.** We test our approach’s ability
 to identify and control the Allegro Hand under heavy appearance perturbations. Recall from Section 5.5 in the
 main paper that, during training, our data augmentation process samples background noise to make the model
 invariant to background perturbations.

During testing, we introduced significant perturbation to the scene’s geometry and appearance by fencing
 off the Allegro Hand with several pieces of white cardboard, shown in **Figure 4a**. We used the 3D trajectory
 tracking scheme described above to control the Allegro Hand to close. We report the quantitative and qualitative
 results in **Figure 4b,c**. We find that our method is able to control the Allegro Hand to follow 3D trajectories in
 the face of scene perturbations. We numerically find that our approach can obtain high-quality tracking results,
 achieving a median joint error of 2.89 degrees.

Figure 5: **Additional evaluations on scene perturbation.** **a**, We conduct additional evaluations on the robustness of our method against scene geometry and appearance perturbations. We placed several pieces of cardboard to fence off the hand, perturbing the visual scene. We run the reference trajectory eight times **b**, We visualize the predicted 3D state of our method as it controls the Allegro Hand to follow a 3D trajectory. **c**, We run each trajectory eight times to compute the mean and standard deviation of the errors, measured as the joint angle differences between the achieved end state and the final tracking goal.

4.4 (R2) Investigation 4: Evaluations of the learned Jacobian in comparison with physics simulators.

R2: "For some systems, one could analytically infer the jacobians and report a measure of how well the learned model captures these."

 We conducted experiments comparing our learned Jacobain model against analytical models for the Allegro
 Hand and the HSA Platform. In **Figure 4**, we created a violin plot that numerically assesses the quality of our
 predicted Jacobian fields against the analytical counterpart. Our quantitative results reaffirm that our model can
 reconstruct high-quality, interpretable physical quantities by learning from visual observations.

**Allegro Hand** Using Drake [25], a state-of-the-art rigid robot simulation, we quantitatively found that our
 model performs well in predicting the angles of the Jacobian vector for each command channel, resulting in an
 average error of 7 degrees. Qualitatively, we visualize our comparison against the analytical model in **Figure 7**.

**Experimental Procedure.** We sample 3D points inside the collision volume of the Allegro Hand in Drake.
 We analytically compute the transformed Jacobian at the queried point by extending the kinematic tree. For
 our Jacobian field, we similar sample 3D points inside the scene uniformly using a 3D grid. Using Procrustes
 analysis [14], we compute the 3D transformation that registers the analytical Jacobian point cloud to our predicted
 Jacobian point cloud. For every point in the predicted Jacobian field, we find the five nearest neighboring points
 from the analytical Jacobian field and take their average Jacobian matrix. We compare the angle differences of
 each command-channel vector inside between the two Jacobian matrices.

**Handed Shearing Auxetics Platform** Using the analytical model of the HSA platform proposed in [23],
 we compare our learned model by using an expert-designed model as the oracle. The analytical model of the HSA
 platform from [23] is a 2D model that only supports 2D motion. Notably, it does not account for motions that
 are inherently 3D, such as the twisting motion that the HSA platform is capable of. The model uses a discrete
 Cosserat approach from [21] and assumes that the backbone of each HSA actuator maintains a constant strain. To
 make it amenable to our formulation, we adopt a quasi-static approximation to map motor actuator commands to
 their corresponding state variables. However, this model struggles to capture the dynamics accurately in straight
 configurations due to a singularity encountered when the HSA actuators are straight, as the curvature of the HSA
 backbone becomes zero. Therefore, we evaluate their model in different 2D configurations where the HSA is bent.

We emphasize that the expert-designed model is not a perfect reflection of the physical reality of the HSA
 system. It uses simplified 2D geometry and does not model or support 3D twisting motions. To the best of our
 knowledge, our model is the only current solution for modeling the 3D twisting+tilting dynamics of the HSA
 platforms. The current best expert-designed models can only support 2D tilting motions.

**Experimental Procedure.** Since the analytical model only supports 2D HSA configurations, where the HSA
 legs on the two sides are symmetrical, we command the HSA platform to compliant testing configurations. For
 our Jacobian field, we sample 3D points inside the scene to obtain the predicted Jacobian point cloud. We project
 the 3D point cloud to 2D and use Procrustes analysis [14] to align with the coordinate system of the 2D analytical
 model. For every point in the predicted Jacobian field, we find the five nearest neighboring points from the
 analytical Jacobian field and take their average Jacobian matrix. We compare the angle differences of each
 command-channel vector inside between the two Jacobian matrices.

Figure 6: **Comparison against analytical Jacobian models.** We compare our Jacobian predictions with analytical counterparts computed via physics simulation [25, 23]. Our method learns consistent Jacobian measurements from raw RGB observations, achieving high-quality median Jacobian errors on both robotic systems.

Figure 7: **Visual Jacobian comparison against analytical simulations.** **a**, We qualitatively evaluate the quality of our Jacobian predictions by visualizing the analytically computed Jacobian field for the Allegro Hand in Drake [25]. We find that our model highly matches the analytical Jacobian field, including the low-to-high opacity transition on each finger, which indicates the total change of position upon a command. We highlight that computing the analytical Jacobian relies on the existence of a physics simulator and a URDF file. **b**, We surgically compare each command component of the predicted Jacobian (red), a 3D vector, queried at a 3D position in the Jacobian field. The analytical result is drawn as a blue arrow. Each column visualizes a different command channel inside the Jacobian, covering the entire space of robot motors. As evidenced by the results, our model achieves high-quality Jacobian predictions, learning from just video observations.

	Thumb	First Finger	Middle Finger	Little Finger
Error Mean (deg.)	10.971	8.651	5.581	6.323
Error Std (deg.)	0.215	0.178	0.239	0.395

Table 3: **Quantitative comparison against analytical Jacobian (physics-simulation)**. We assess the quality of the Jacobian predictions made by our learned neural network by comparing our model against the analytical Jacobians computed by a state-of-the-art physics simulation Drake [25]. We used five testing samples representing different hand poses and reported the mean and standard deviation of the errors.

5 Author’s Response to Reviewer 1

5.1 “First of all, the idea of using optical flow to parameterize dynamics models is not
new... Most of the key claimed properties—such as self-supervised learning of
dynamics models—apply to these prior systems as well.”

We thank the reviewer for bringing up prior work that has parameterized dynamics models with the flow
as the output. We have added a related work section to our manuscript **Sec. A.12**, addressing the differences
between our work and these works.

- 1. We agree with the reviewer that using optical flow as the supervision signal is not the key novelty of our
work. We also did not claim that as a novelty in our submission.
- 2. We would like to clarify the key novelty of our work:
 - (a) First, as Reviewer 1 and Reviewer 2 have kindly mentioned, **one key novel insight from our work is**
**that many robots have difficult-to-model system dynamics**. This has been a persistent challenge
for low-cost and soft robots, where the forward dynamics cannot be analytically modeled. We
investigated this challenge and provided a vision-based solution. Our solution is validated across a
**diverse range of robotic systems** that can learn **controllable and interpretable** system dynamics.
 - (b) As Reviewer 2 mentions, the second novel insight from our work is that the 3D system Jacobian of
robot surfaces can be learned in a self-supervised manner from 2D images in an end-to-end fashion.
 - (c) The Jacobian Field parameterization for kinematics and geometry itself is a key novelty. As highlighted
in our experiments in **Investigation 2 (Sec. 4.2)**, this design dramatically improves out-of-distribution
generalization of the flow predictor by exploiting locality and linearity, critical for the low-data regime
of robotics.
- 3. To the best of our knowledge, **there has not been a practical solution to the automatic inference**
**of the 3D volumetric system dynamics of diverse robotic systems** from vision alone. **Our method is**
**novel in the sense that prior work cannot be applied to solve the same problem.**
- 4. Lastly, our method is heavily inspired by challenges and opportunities across new manufacturing
techniques (e.g., multi-material ink-jetting can scalably produce affordable robotic systems without
assembly, but these systems are challenging to model) low-cost robotics automation (e.g., a 3D-printed
or self-assembled robot arm that is affordable and closed-loop controllable but comes with unreliable
embedded sensors). We note that our problem statement is thus deeply practical - we are making a step
towards solving a real problem that has, to date, held back the deployment of these systems.

5.2 “Yet strangely, it emphasizes an issue where this is insufficient: backlash... However,
the amount backlash depends on historical actions and forces as well. The paper
restricts the setting in a rather non-transparent way, saying in the appendix that “In
practice, our nominal point represents a steady state, as one can wait for the robot
command to settle.”—which is simply not true for any task involving unstable
manipulation or where speed matters.”

Thank you for bringing up the discussion of backlash and the values of modeling second-order transients!

**Clarification on backlash.** Following your review, we found that the term “backlash” is overloaded and
used by different communities to refer to different phenomena. Some communities use the term backlash in the
way you understand it - as second-order transients of a physical system.

However, backlash is also commonly used in the mechanical engineering and robotics communities to refer to a
situation where a joint “wiggles”, i.e., has play or slop, due to low-precision manufacturing, such as gears that are
not fitting snugly [5, 18, 22]. In this case, backlash leads to situations where joint sensors or proprioception of the
robot report joint angles that are inaccurate and are not indicative of the true 3D configuration of the robot [9].
Taking the DIY Poppy robot arm as an example, backlash causes the joint sensors to report the same values for
different 3D configurations of the robot (**Please see Figure 1 of Ref. [46] in our paper for more details**). In

the paper, we showed that visual observations can resolve this issue by measuring the true extrinsic state of the
system. When our approach is employed in closed-loop, as demonstrated in the results, we can command the
Poppy DIY arm, a system experiencing backlash, to perform trajectory following accurately.

**In the revised Methods Section A.2**, we clarified the definition of backlash explicitly in the paper and added
references that specify this definition further. We clarify explicitly that we *do not claim* that we are modeling
second-order transients. We agree with the reviewer that modeling second-order transients, such as acceleration
and damping, can further reflect the true physical system we consider in this work.

*Is it necessary to model second-order transients?* While we agree with the reviewers that certain
manipulation tasks could require modeling second-order transients, many prior works in manipulation planning
have suggested that quasi-static models can hold up for a wide variety of manipulation tasks, including many
that involve dexterous hands [19]. These models only capture physical state transitions between equilibria. We
therefore have followed prior works and decided to build a quasi-static model. As [19] has shown, although
modeling the transients allows the discovery of more dynamic behaviors, the added computational complexity
frequently outweighs the benefits. Our results align with prior works on these perspectives, as we have found
that modeling transitions between static equilibria can enable control of diverse and complex robotic systems.
Different from these prior works that assume a physical description of the system is provided by an expert, our
work shows that the linearization between steady states can be used to learn volumetric Jacobians via visual
observations. **We added a discussion of modeling assumptions discussion in a new Methods Section A.6.** In
the main text, we revised our discussion section (**Sec. 4 of our paper**) to refer to **Sec. A.6**.

We thank the reviewer for bringing up the backlash issue and the value of considering second-order transients.
Modeling second-order transients is compatible with our Jacobian approach through modeling additional constraint
forces [1, 2, 19], because due to conservation of energy the same Jacobian relationship holds between joint
torques and spatial forces [16]. We think this is an exciting next step for our research project. **We also revised the**
**caption of Figure 3 in our paper to make clear that we are not claiming to model second-order transients.**

**5.3 “The model computes Jacobians based on 1) instantaneous appearance of the object
in the camera, and 2) 3D position obtained from depth cameras.”**

We would like to clarify that our model does not take 3D positions obtained from depth cameras to compute
the Jacobian values. During training, our model uses RGBD observations from multiple views as supervision to
learn the appearance, geometry, and kinematics of the robotic system. During testing, only one single RGB camera
is needed to infer the 3D representations that contain appearance, geometry, and kinematics information - no
depth is used at inference time.

**5.4 Differences between our work and prior work in parameterizing dynamics models
with 3D scene flow or 2D optical flow.**

1. Perhaps most relevant is Xu et al. 2020, "Learning 3D Dynamic Scene Representations for Robot Manipulation," which also parameterizes a dynamics model using 3D scene flow, and does it not just for the robot, but also for the rest of the scene.
2. Argus et al. 2020 "FlowControl: Optical Flow Based Visual Servoing" shows that optical flow-based dynamics models can be used for more complex control problems than those shown here (i.e. manipulation).
3. And similarly Lee et al. "Aggressive Perception-Aware Navigation using Deep Optical Flow Dynamics and PixelMPC" shows flow-based dynamics models can be used for planning drone movement.

We thank the reviewer for their suggestions on a list of related works. We have clarified our differences from
Xu et al. in **Clarification 4 in (Sec. 2 of this document)**. For all listed works, we have addressed our differences
in **the related work section (Sec. A.12) in our revised paper**.

**5.5 “Moreover, the inclusion of 3D position as an input to the dynamics model seems
strange, as it means that the dynamics model may break if the entire robot is moved
in 3D.”**

**About the inclusion of 3D positions.** We thank the reviewer for the constructive feedback. Including 3D
positions in the model is necessary for controlling unobserved parts. It is the key for use cases demonstrated in
our 3D trajectory tracking experiments (Fig. 1). Please see our **Investigation 1** in **Section 4.1** of this document
for more details. 3D modeling is also a natural choice that reflects the reality and mechanics of robotic systems in
our 3D physical world.

**About moving the robot in 3D.** We humbly acknowledge that moving robots in 3D brings challenges to
the identification and perception of the system. This is discussed in **Investigation 3 (Sec. 4.3)**. **Our results in**
**the original submission also show our model’s robustness against moving robots in 3D.** Our training dataset
for the pneumatic hand does not cover the robot arm configuration in the testing scenarios. We illustrate this
out-of-distribution generalization capability in **Figure 8**, where we overlay the training dataset hand motions on
top of the testing configurations.

Figure 8: **Out-of-distribution Generalization.** We visualize all point tracks related to finger motions in the Pneumatic Hand dataset. We overlay these point tracks on a testing sample on the left and a training sample on the right. The testing overlay shows that the Pneumatic Hand is clearly out of training distribution at test time. In Figure 3 of our manuscript, we show that our method is able to control the fingers to close and grasp the spatula. This indicates that our dynamics model is robust when the robot hand locally moves in 3D.

5.6 “no experiments are done where the grippers are on the end of an arm, probably because this will cause problems for the model. The highly restrictive inputs are likely required, however, in order to maintain invariance to distractors, as the model can use the 3D position as a relatively unambiguous cue for the robot’s overall pose. As such, I expect the neural network will likely not generalize beyond the scenes it has been trained on.”

We thank the reviewer for their insightful comment on generalization over changing morphology and camera
settings.

**Open Challenges.** We humbly acknowledge that the following settings remain open challenges:
Training on a standalone robot hand placed on a table and expecting the model to generalize when the robot hand
is mounted onto an arm with a changing camera viewpoint. Our model might fail to generalize at inference time
if the spatial and geometric relationship between the single camera and the robot is severely out of distribution.
In our initial manuscript, these challenges are addressed in a pragmatic spirit – training the model after the
hand is mounted on the robot arm. In **Investigation 3 (Sec. 4.3)** and the new **Methods section A.9** in our
**revised paper**, we humbly acknowledge the open challenges to our system due to changing camera and scene
configurations. We then provide new results and analyses on the generalization capability of our model.

**Clarification of the capability of our model.** In fact, experiments in our original submission already
cover the case where a gripper is mounted on the end of an arm. As we have shown in **Figure 3 of the original**
**manuscript**, we could train our system to jointly control the pneumatic hand and the UR5 arm. This way, the
Jacobian field can describe the motions of both the hand and the arm. We have shown that this solution can, in
practice, enable many quasistatic manipulation tasks, such as planar pushing and object grasping. In our paper,
we assume that the relative transformation between the robot and the single-camera used at inference time will

be within the training distribution. **Given modern engineering standards for camera and robot placements,**
**this is a scalable practice since these constraints only concern the single RGB camera during inference.**

Lastly, our work is particularly well-positioned to inspire future research to address these important challenges.
We deeply thank both Reviewer 1 and Reviewer 2 for their questions on generalization. (E.g., Reviewer 2: “If a
model is learned in one lab, what all needs to be done to use it in a different lab setup”). As Reviewer 2 has kindly
mentioned, “the work presented here can be a great stepping stone to eventual solutions which resolve these
issues in practical deployment.”

5.7 “The paper also broadly ignores the fact that local appearance has problems with domain shift: for example, if the robot is manipulating a previously-unseen object, jacobians will change as the gripper comes into contact with the object, and it’s unlikely that the neural net will correctly model this due to the reliance on appearance. Some experiments are done to address robustness (fig. 4 a,b), but these more argue that incorrect Jacobians are sufficient in this particular instance, rather than showing that the model can estimate correct Jacobians under domain shift.”

About the impacts of environment variables and second-order transients. You are right that the dynamics will change if non-robot environment variables or second-order transients dominate the system dynamics. I.e., putting a rubber band onto the robot finger at inference time or cutting a hole on the pneumatic hand with a scissor. **Although our system can plan robot motions under changing dynamics, our system does not explicitly model the dynamics of changing environment variables, such as the elasticity of an unseen rubber band and unseen objects.**

About Jacobian generalization. We empirically find that our system is still highly useful for a wide range of manipulation tasks, such as controlling the pneumatic hand and the arm to push and grasp objects and rotating and bending the HSA system with an unseen weight appended. The incorrect Jacobians are sufficient in these cases because the predicted Jacobians’ directions remain largely correct, though their scale does not, which is effectively addressed by closed-loop control. Our system tackles this out-of-distribution challenge successfully.

Our system is compatible with second-order transient modeling. Physically speaking, putting the rubber band on the robot finger will add constraint forces, but locally, the rubber band does not change the directions of the system Jacobians. If constraint forces from contacts and environment interactions are modeled, by summing up the influences of the joint forces and the constraint forces, one should be able to capture the reality of the physical system more accurately. Hence, our model is compatible with modeling a higher degree of physical fidelity.

We think extensively addressing both robot dynamics modeling together **with environment dynamics modeling in a single manuscript is beyond the scope of our work and would dilute the focus of our investigation**, especially in the aspects of novel manufacturing techniques and low-cost 3D printed robots. However, our proposed Jacobian parameterization is highly expressive in mechanical terms and can be extended to work with non-robot environment variables and second-order transients, as explained above. We especially note that the *same* system Jacobian could in the future be used to model second-order transients and characterize relationships between spatial force and joint torques by the virtue of energy conservation [16].

5.8 “The writing of the methods section emphasizes the NeRF aspect, but the experiments do not make it clear if this is actually necessary or even useful. Xu et al. achieves similar dynamics learning from RGB-D and scene flow. Why should this paper be any different? I.e., why not just use standard scene flow to estimate 3D motion directly, and regress to that? This would remove the reliance on multiple cameras. “

De-emphasizing NeRF [17] and Differentiable Rendering. In **Proposed Writing Change 4 in Sec. 3 of this document**, we present our efforts on de-emphasizing NeRF and Differentiable Rendering in the revised paper. We have made several editorial changes and added a new Methods section discussing alternatives in the supervision of our proposed Jacobian fields, which suggests a future baseline without NeRF for the readers.

*Differences from Xu et al. (2020).* We agree that Xu et al. is an excellent prior work. Our **Investigation 2**
**(Sec. 4.2 of this document)** and **Clarification 4 (Sec. 2 of this document)** provide discussions of our differences
from Xu et al. (2020). Our work is different regarding problem setting, methodology, and capabilities.

*Comparison with direct 3D scene flow prediction.* In **Investigation 2 (Sec. 4.2)**, we included an
extensive set of experiments justifying the usefulness of our Jacobian parameterization. We experimentally found
in both real-world and simulated datasets that our Jacobian parameterization narrows the search space for the
right dynamics model, due to its properties of spatial locality and linearity. We additionally found that our Jacobian
parameterization enables generalization to unseen configurations of robots. The direct flow prediction baseline
fails to generalize in these scenarios.

*Reliance on multiple cameras.* **Our reliance on multiple cameras only exists at training time but not**
**at deployment.** We think the added benefits of 3D modeling and Jacobian parameterization outweigh the burden
of multiple cameras. We anticipate that the need for multiple cameras will be reduced by the advancements of
large image-to-3D foundation models [24, 12] that can be fine-tuned for our application.

5.9 “What’s more, most of the evaluations are done by following 2D point trajectories,
which raises the question of whether 3D is needed at all. Vecerik et al. 2024
"RoboTAP: Tracking Arbitrary Points for Few-Shot Visual Imitation" (also not cited)
achieved 3D control by using 2D visual servoing of dense surface points.”

*The usefulness of 3D.* We deeply thank the reviewer for their constructive suggestion. We hope that our
**Investigation 1 (Sec. 4.1)** and new experiments have provided further justifications for why 3D is useful. Our
investigation contains additional evaluations conducted in 3D, including trajectory tracking in 3D using model
predictive control. **We emphasize that the value of 3D should be seen at training time as well**, as it constrains
the space of possible solutions for the neural dynamics function that explains our motion observations.

*Differences with Vecerik et al. (2024) [28]* We have discussed differences with the elegant work by
Vecerik et al. (2024) [28] in our new related work section in **(Sec. A.12)** in our revised paper.

5.10 “On the other hand, one very novel and, in my opinion, highly under-emphasized
aspect of this paper is the use of novel point-tracking methods [41] as a method of
parameterizing demonstrations. The current paper spends just a single paragraph
positing a solution to this very fundamental problem, stating "The desired 2D
motion is computed as the difference between the target and current 2D locations"
without giving any explanation of where the 2D targets come from other than that
very recent TAP models are used for this. In some cases, the targets appears to be a
final configuration, but in other cases, the model follows an entire trajectory, in
which the target locations must change throughout the episode. There is no
discussion of this in the supplementary. In practice, this 'demonstration following'
for surface points is critical for making Jacobians of surface points a sensible
representation, and is critical for making the whole algorithm useful. Almost no
attention is given to it.”

We are grateful for the commendation of the reviewer. We fully agree that this aspect of our work is novel and
that emphasizing this aspect will strengthen our work further.

**Revised writing.** We strengthened our paper’s connections with 2D trajectory tracking for general imitation
learning. Please see our **Proposed Writing Change 1 in Section 3**. We have improved the focus of our discussion
on using novel point-tracing methods for parameterizing demonstrations.

1. We rewrote a large portion of our method description(**Section 2**) in our revised paper to connect with the
general imitation learning applications that our learned dynamics model can offer (**L99-109, L165-L183**).
Following Reviewer 1’s constructive suggestion, we emphasized the following in our revised texts.

- (a) Our inverse dynamics controller parameterized desired motions densely in 2D image space or 3D to
find robot commands.
- (b) We find that parameterizing demonstration trajectories as dense point motions is the key to controlling
a diverse range of robotic systems, as motions of deformable and dexterous robots cannot be
well-constrained via rigid transformations specified on a single 3D frame.
- (c) Our parameterization enables a wide range of robots to imitate video-based demonstrations.
- 2. Our new related work section in **a new Methods Sec A.12** clarifies our connections with trajectory
tracking and differential kinematics methods.

**Implementation details on trajectory tracking.** We have included pseudo-code for our model predictive
controller in our revised paper, as well as a detailed discussion of demonstration processing, state encoding,
and command optimization in **Sec. A.6 of our revised paper**. We publicly released our implementation under
<https://github.com/sizhe-li/neural-jacobian-field>.

**Presentation details on trajectory tracking and goal states.**

- 1. We appreciate the reviewer for their writing suggestion on experiment presentations.
- 2. We have added clarifications in the supplementary (**Sec. A.6 of our revised paper**). We clarified that some
trajectory-tracking tasks are specified with multiple waypoints, while others (with very short horizons)
are specified with a single waypoint.
- 3. Building a generalist scene-flow-based motion planner is a great future direction. Our work focuses on
the control aspect and models the dynamics of a system. We think that these two will come together in
the future.

6 Author’s Response to Reviewer 2

a) It is unclear what the coordinate system in which a point x is defined. Given an image, are the points x defined in the coordinate space of the camera, or is there an assumed canonical coordinate system? If the latter, does this imply that the learned models would not be easily applicable to slight new setups (e.g. moving the robot arm to a new lab), or would at least require some calibration of identify the new camera position w.r.t. the robot-centered canonical coordinate system?

- 1. We thank the reviewer for the insightful remark. x is defined in the camera coordinate system.
- 2. To your interests, we would like to refer you to our additional results in **Investigation 1** and **Investigation**
**3** in **Sec. 4.1**.

b) The results are primarily qualitative (albeit with some evaluation for how well some desired trajectories are followed). However, for some systems (e.g. the Allegro hand), one could analytically infer the jacobians and report a measure of how well the learned model captures these. Similarly, for some soft robots, an approximate analytical can be defined and a demonstration that the learned model captures the dynamics better (and allows more accurate control) would be helpful in empirically demonstrating the benefits of the approach.

- 1. Thank you for your suggestions that strengthen our work! We conducted experiments comparing our
learned Jacobain model against analytical models of the Allegro Hand and the HSA Platform.
- 2. Please see our **Investigation 4 in Section 4.4 of this document** for detailed results and analysis of our
new comparison against the analytical model.
- 3. We have added our new results and analysis into **a new Methods Section A.10** in the revised paper
- 4. We emphasize that the expert-designed model is not a perfect reflection of the physical reality of the HSA
system. It uses simplified 2D geometry and does not model or support 3D twisting motions.

c) The self-supervised learning setup relies on giving random motor commands and observing its effects. However, if the learned model is to be generally useful, it should observe training states across varying configurations of the robot (e.g. different combinations of open fingers). Are random motor commands really sufficient to yield such a diverse distribution? Or is an exploration strategy required to ensure that the robot configuration space can be adequately covered?

1. Thank you so much for the constructive suggestions! You are right that the dynamics model needs to observe training states across varying robot configurations.
2. Random commands were sufficient for the robotic systems we covered in this study. For the Allegro Hand, we initialized the starting configuration of each trajectory in the dataset using a sparse grid of joint values; the subsequent commands in each trajectory are still randomly sampled.
3. We agree with the reviewer that an exploration strategy will be useful for systems with combinatorial complexity, such as a more complex dexterous robot hand (>20 DoFs) mounted on a robot arm. We think this is an exciting future direction for our research and would connect with prior works in curiosity learning [20]. Given the scope of our work, this was not the top focus of our study.

d) The neural network is based on PixelNeRF, which uses pixel-aligned features from the input image to inform prediction for any 3D point. However, a common limitation of such a system is that the information for occluded 3D points is still based on visible ones before them (e.g. the radiance fields learned by pixelNeRF in occluded regions are typically blurry) and this may reduce the accuracy of learned dynamics for hidden 3D points. As an alternative, could a single (image-encoded) global latent-conditioned neural field be a more robust solution?

1. Thank you for the insightful suggestion! While it is true that pixelNeRF uses pixel-aligned features, these features are nevertheless a function of the *full* image, as the residual network encoder has a receptive field that significantly exceeds the full image.
2. Nevertheless, you are correct in that better global information would likely boost performance further. This could be effectively addressed with transformer-based visual encoding (DINO [6]), which contains a global token that can be trained jointly with the pixel-aligned decoder. A future iteration of our method would build on such a more powerful encoder with a better global context.
3. An ideal approach would likely, as you suggest, *combine* global and local conditioning - we are excited to explore this in future work.
4. Even in this case, however, occluded surfaces would generally remain blurry up to the degree of *uncertainty*, that is, given visual observations, the model *cannot* be certain what the occluded surface looks like. As our present model is a deterministic model, this uncertainty leads to predicting the mean of all possible reconstructions, which is blurry. We have explored and solved this problem in prior work which builds *probabilistic conditional generative models* based on pixelNeRF [27]. Our proposed method could be built on this framework as well, which would effectively address blurriness in occluded parts of the image and effectively address uncertainty.
5. As evidenced by **Extended Data Figure A4 in our manuscript**, though blurry, reconstructions obtained by our method remain good enough to obtain control of occluded surfaces with our proposed Jacobian Field in practical scenarios, though methods discussed above would likely boost performance further. Thank you for your valuable suggestions!
6. We added a detailed discussion on the limitation of PixelNeRF and the importance of global information in **the new Methods Sec. A.2** in the revised paper.

f) Perhaps “Neural Jacobian Field” is too broad a term — this models jacobian of 3D point motion w.r.t motion commands, but there are other jacobians one could associate with points in space. In fact, a prior with with a similar name already exists (“Neural Jacobian Fields: Learning Intrinsic Mappings of Arbitrary Meshes, Aigerman et. al.”), using jacobians to model mesh manifold. Perhaps some more specific name maybe better that captures to robotics-related nature of the learned fields “Neural Actuation Fields” (not a binding suggestion!)?

Thank you for your suggestions on our method name (initially Neural Jacobian Fields). As clarified in **Sec. 3 of this document**, we propose “Visuomotor Jacobian Fields.” We have this proposal in our revised paper.

References

- [1] S. Andrews, K. Erleben, and Z. Ferguson. Contact and friction simulation for computer graphics. In *ACM SIGGRAPH 2022 Courses*, pages 1–172. 2022.
- [2] A. W. Bargteil, T. Shinar, and P. G. Kry. An introduction to physics-based animation. In *SIGGRAPH Asia 2020 Courses*, pages 1–57. 2020.
- [3] J. Bonet and R. D. Wood. *Nonlinear continuum mechanics for finite element analysis*. Cambridge university press, 1997.
- [4] R. S. Bowen, R. Tucker, R. Zabih, and N. Snavely. Dimensions of motion: Monocular prediction through flow subspaces. In *2022 International Conference on 3D Vision (3DV)*, pages 454–464. IEEE, 2022.
- [5] R. Budynas and J. K. Nisbett. *Shigley’s Mechanical Engineering Design*, volume New York. McGraw-Hill New York, 2015.
- [6] M. Caron, H. Touvron, I. Misra, H. Jégou, J. Mairal, P. Bojanowski, and A. Joulin. Emerging properties in self-supervised vision transformers, 2021. URL <https://arxiv.org/abs/2104.14294>.
- [7] D. Charatan, S. Li, A. Tagliasacchi, and V. Sitzmann. pixelsplat: 3d gaussian splats from image pairs for scalable generalizable 3d reconstruction. In *CVPR*, 2024.
- [8] C. Chi, S. Feng, Y. Du, Z. Xu, E. Cousineau, B. Burchfiel, and S. Song. Diffusion policy: Visuomotor policy learning via action diffusion. In *Proceedings of Robotics: Science and Systems (RSS)*, 2023.
- [9] J. J. Craig. *Introduction to Robotics: Mechanics and Control, 3rd Edition*. Pearson, 2004.
- [10] M. Cuturi. Sinkhorn distances: Lightspeed computation of optimal transportation distances, 2013. URL <https://arxiv.org/abs/1306.0895>.
- [11] J. Feydy, T. Séjourné, F.-X. Vialard, S.-i. Amari, A. Trounev, and G. Peyré. Interpolating between optimal transport and mmd using sinkhorn divergences. In *The 22nd International Conference on Artificial Intelligence and Statistics*, pages 2681–2690. PMLR, 2019.
- [12] Y. Hong, K. Zhang, J. Gu, S. Bi, Y. Zhou, D. Liu, F. Liu, K. Sunkavalli, T. Bui, and H. Tan. Lrm: Large reconstruction model for single image to 3d. *ICLR*, 2024.
- [13] D. Koschier, J. Bender, B. Solenthaler, and M. Teschner. Smoothed particle hydrodynamics techniques for the physics based simulation of fluids and solids. *arXiv preprint arXiv:2009.06944*, 2020.
- [14] W. J. Krzanowski. *Principles of multivariate analysis: a user’s perspective*. Oxford University Press, Inc., USA, 1988. ISBN 0198522118.
- [15] Z. Li, T. Müller, A. Evans, R. H. Taylor, M. Unberath, M.-Y. Liu, and C.-H. Lin. Neuralangelo: High-fidelity neural surface reconstruction, 2023. URL <https://arxiv.org/abs/2306.03092>.
- [16] K. Lynch. *Modern Robotics*. Cambridge University Press, 2017.
- [17] B. Mildenhall, P. P. Srinivasan, M. Tancik, J. T. Barron, R. Ramamoorthi, and R. Ng. Nerf: Representing scenes as neural radiance fields for view synthesis, 2020. URL <https://arxiv.org/abs/2003.08934>.
- [18] R. L. Norton. *Design of Machinery: An Introduction to the Synthesis and Analysis of Mechanisms and Machines, Third Edition*. McGraw-Hill, 2004.
- [19] T. Pang, H. J. T. Suh, L. Yang, and R. Tedrake. Global planning for contact-rich manipulation via local smoothing of quasi-dynamic contact models, 2023. URL <https://arxiv.org/abs/2206.10787>.

- [20] D. Pathak, P. Agrawal, A. A. Efros, and T. Darrell. Curiosity-driven exploration by self-supervised prediction. In
*International conference on machine learning*, pages 2778–2787. PMLR, 2017.
- [21] F. Renda, V. Cacucciolo, J. Dias, and L. Seneviratne. Discrete cosserat approach for soft robot dynamics: A new piece-wise
constant strain model with torsion and shears. In *2016 IEEE/RSJ International Conference on Intelligent Robots and*
*Systems (IROS)*, pages 5495–5502. IEEE, 2016.
- [22] B. Siciliano, S. Lorenzo, L. Villani, and G. Orilo. *Robotics: Modelling, Planning and Control (2nd edition)*. Springer
London, 2010.
- [23] M. Stölzle, D. Rus, and C. Della Santina. An experimental study of model-based control for planar handed shearing
auxetics robots. In *International Symposium on Experimental Robotics*, pages 153–167. Springer, 2023.
- [24] J. Tang, Z. Chen, X. Chen, T. Wang, G. Zeng, and Z. Liu. Lgm: Large multi-view gaussian model for high-resolution 3d
content creation, 2024. URL <https://arxiv.org/abs/2402.05054>.
- [25] R. Tedrake and the Drake Development Team. Drake: Model-based design and verification for robotics, 2019. URL
<https://drake.mit.edu>.
- [26] Z. Teed and J. Deng. Raft: Recurrent all-pairs field transforms for optical flow, 2020. URL [https://arxiv.org/abs/](https://arxiv.org/abs/2003.12039)
[2003.12039](https://arxiv.org/abs/2003.12039).
- [27] A. Tewari, T. Yin, G. Cazenavette, S. Rezchikov, J. B. Tenenbaum, F. Durand, W. T. Freeman, and V. Sitzmann. Diffusion
with forward models: Solving stochastic inverse problems without direct supervision. In *Conference on Neural Information*
*Processing Systems (NeurIPS)*, 2023.
- [28] M. Vecerik, C. Doersch, Y. Yang, T. Davchev, Y. Aytar, G. Zhou, R. Hadsell, L. Agapito, and J. Scholz. Robotap: Tracking
arbitrary points for few-shot visual imitation. In *ICRA*, pages 5397–5403. IEEE, 2024.
- [29] P. Wang, L. Liu, Y. Liu, C. Theobalt, T. Komura, and W. Wang. Neus: Learning neural implicit surfaces by volume
rendering for multi-view reconstruction, 2023. URL <https://arxiv.org/abs/2106.10689>.
- [30] C. Wen, X. Lin, J. So, K. Chen, Q. Dou, Y. Gao, and P. Abbeel. Any-point trajectory modeling for policy learning, 2024.
URL <https://arxiv.org/abs/2401.00025>.
- [31] Z. Xu, Z. He, J. Wu, and S. Song. Learning 3d dynamic scene representations for robot manipulation, 2020. URL
<https://arxiv.org/abs/2011.01968>.
- [32] X. Yinghao, S. Zifan, Y. Wang, C. Hansheng, Y. Ceyuan, P. Sida, S. Yujun, and W. Gordon. Grm: Large gaussian
reconstruction model for efficient 3d reconstruction and generation, 2024.
- [33] A. Yu, V. Ye, M. Tancik, and A. Kanazawa. pixelNeRF: Neural radiance fields from one or few images. In *CVPR*, 2021.

Point-by-point Response to Reviewers

General Response

Dear Reviewers,

We sincerely thank the editors and reviewers for the fruitful discussion throughout the whole peer-review process,
which helped us strengthen our work. We are pleased that the reviewers found the revised manuscript adequate
in addressing their suggestions.

In the following, we summarize changes we made to the manuscript addressing final reviewer comments.

Sincerely,

*On behalf of all authors,*

Vincent Sitzmann

1 Author's Response to Reviewer 1

1.1 Concerns related to changing orientations of robots.

"I apologize for mis-stating that there were no experiments on robot arms. There is one such example, but even here the use of the the arm is limited, and overall it remains problematic for the same reasons. Specifically, from the experiments shown, it appears that the robot arm/wrist has only 2 degrees of freedom, with most of the degrees of freedom in the hand. The result is that if a particular point on the robot's surface is at a particular 3D location, then there is very little remaining ambiguity about its Jacobian. This is still far from more practical situations where a robot hand can move its arm/wrist in a full 6 degrees of freedom. In such scenarios, even if a given point on the robot's surface is at a particular location, it may be at a different *orientation*. This, therefore, puts a much heavier burden on the vision part of the network to estimate the robot's 3D pose, as different 3D poses for the same point may have different Jacobians. This kind of setup is common even in warehouse pick-and-place tasks, where obstacles might mean that the robot needs to reach to the same object from a different orientation. The current paper admits this weakness to some extent, but states it as "Relocating the camera to an arbitrary viewpoint in the scene and generalizing to that view." This feels somewhat misleading, as readers may think that this will work for any application with a static camera, when in fact the algorithm will have the same problem if any of the robots embodiments shown here are put on the end of a long arm with 6 degrees of freedom that is free to move arbitrarily through the environment."

*About unseen robot poses.* We agree with the reviewer that robot configurations unseen during training
pose a challenge for any method that relies on deep neural networks to infer robot state from vision.

While our approach is not an exception—and robot poses that fall far outside the training distribution do
remain difficult—our system offers several advantageous properties. First, it is fully data-driven, meaning that
one can simply collect additional data for underrepresented robot poses. Second, experimental evidence already
demonstrates generalization of our system to correctly predicting Jacobians for robot poses *unseen* during training
time. We observe this on the pneumatic hand, where we can predict correct Jacobians even for a location of the
hand that is unseen in the training set. The compositional nature of parameterizing robot motion via the linear

Jacobian, highlighted in both real-world experiments (**Allegro Hand; Extended Data Fig. 5**) and simulated
settings (**2D Finger Environment; Fig. 2, Supplementary Information**), serves to further reduce the amount of
data necessary for accurate predictions.

We fully agree with the reviewer: "it also seems plausible that more data will lead to stronger generalization,
i.e., the model will eventually learn good jacobians regardless of 3D location by learning a good appearance
model.". We believe that this is a particular strength of our method: it is relatively straightforward to collect more
data, as no expert intervention is required in the data collection process. Further, we believe that fine-tuning a
vision foundation model to generalize Jacobian predictions *across* embodiments is a particularly promising avenue
towards general-purpose vision-based control.

**Clarification on the combinatorial complexity challenge.** We thank the reviewer again for bringing up
the combinatorial complexity challenge. We believe our approach is a step towards addressing this challenge and
can inspire future research in this direction. In Section A.9, under "open challenges", we have explicitly mentioned
the challenge of predicting Jacobian quantities for robot configurations unseen during training, using the example
Reviewer 1 provided.

1.2 Concerns related to clarifying the contribution of the NeRF.

"My other remaining concern is that, while I appreciate the inclusion of table A4 and I think it strengthens the paper's argument. However, it is also a bit of a strawman. The rebuttal seems most concerned with defending regressing to Jacobians (e.g. "This is in stark contrast to parameterizing f as a neural network that directly predicts scene flow given an image and a robot command since the neural network does not model these symmetries and will thus require orders of magnitude more motion observations to adequately model the system dynamics."), but I'm mostly concerned with clarifying the contribution of the NeRF. The advantages of the Jacobian formulation are analytically clear; I was well aware of the math presented in Observation 1. My main point is that we can compute depth/scene flow at training time (possibly with a NeRF, or with simple depth+flow) and then train a non-nerf model with similar properties to the one here. A sensible ablation, therefore, is to have this model regress per-pixel Jacobians directly from pixels. As such, I would appreciate an additional baseline in table A4 which regresses Jacobians using the same losses as for the rest of the framework. I would also appreciate if this is done on the pneumatic hand rather than the Allegro, since for the Allegro the 3D coordinate almost determines the Jacobian, requiring very little appearance information to disambiguate."

**About the contribution of the NeRF.** We thank the reviewer for their diligence in checking the contribution
of the NeRF. We agree with the reviewer that our proposed Jacobian parameterization is a contribution independent
of NeRF, and valuable in its own right. We already addressed the contribution of NeRF in our previous response.
We emphasized that the core idea of our approach is the field parameterization of the system Jacobian.

Specifically, we made the independence relationship explicit in the Methods section. We clarified that "while
differentiable rendering is a convenient source of obtaining 3D supervision, it is not the only one. If high-quality
depth cameras or motion capture are available, they can also provide 3D information."

However, in implementing the reviewer's proposed baseline, we discovered an unforeseen further advantage
of the 3D field parameterization of the Jacobian.

**Additional baseline comparison.** We sincerely appreciate the reviewer's insightful suggestion on
conducting a baseline that directly regresses the per-pixel Jacobian from pixels. We have followed the reviewer's
comments and conducted the suggested ablation on the pneumatic hand environment. In our experiments, we
removed the neural rendering part of Jacobian Fields by assuming that the model is given depth as input. Keeping
the rest of the architecture unchanged, we train a pixel-aligned 3D Jacobian Fields model as suggested by the
reviewer. This model is supervised using the same losses on depth and flow. Specifically, it needs to predict the
correct 3D Jacobian at each pixel so that the resulting 3D scene flow, when projected to 2D, matches the observed
optical flow. We report quantitative results in **Table 1**, and provide qualitative results in **Figure 1**. We find that this
field-free 3D Jacobian baseline performs *significantly worse*. Specifically, the baseline regularly fails to disentangle

the sensitivities of different surface points to different actuators, predicting incorrect Jacobians. We hypothesize
 that this is due to a lack of multi-view supervision: Without a NeRF that allows us to easily check for which parts
 of the robot are occluded in which view at training time, supervision for the 3D scene flow stems only from the
 same camera perspective that is fed to the image encoder. While, as the reviewer remarks, a NeRF could be used
 at training time to perform occlusion checks, we argue that this approach has little benefit over our proposed
 approach: pixelNeRF does not have major disadvantages over the reviewer’s proposed baseline at test time, as it
 already enables real-time, closed-loop vision-based control, while also providing a full 3D reconstruction including
 occluded parts of the scene. Finally, while alternative modes for obtaining multi-view scene flow supervision
 are possible, such as leveraging ground-truth depth for depth tests to ascertain which 3D points are visible from
 which camera, these significantly increase the complexity of the training pipeline and require hand-tuning of
 hyperparameters. These challenges are well-known in the image-based rendering community, where depth-testing
 was a classic method for inferring which 3D points are visible from which viewpoints, but have largely been
 obviated by differentiable-rendering novel view synthesis. Finally, recent 3D gaussian splatting based approaches
 may offer a sweet spot between depth and neural radiance fields, and are a promising avenue for future work.

We will include the qualitative and quantitative results of the proposed baseline in the supplemental material
 for the paper. We thank the reviewer for this suggestion, which we believe has highlighted an interesting design
 space of interest for future work! We have added our ablative experiment table and figure to the supplementary
 information file.

Optical flow error (pix.)	Mean	Std
Jacobian Field + Depth Input	1.161	1.650
Jacobian Field + Neural Rendering	0.155	0.053

Table 1: **Ablation on neural rendering on the pneumatic hand.** We removed the neural rendering part of Jacobian Fields by assuming that the model is given depth as input. Keeping the rest of the architecture unchanged, we train a pixel-aligned 3D Jacobian model. The model performs significantly worse numerically. Qualitatively, we found that this model fails to disentangle the sensitivities of surface points to different actuators, predicting incorrect Jacobians that over-explain the data. We hypothesize that this is due to a lack of multi-view supervision.

71 **2 Author’s Response to Reviewer 2**

72 Thank you very much for your feedback during the revision phase! We are pleased that you found our rebuttal
 73 and the updated manuscript adequate to address your concerns.

Figure 1: **Ablation on neural rendering on the Allegro Hand.** We removed the neural rendering part of Jacobian Fields by assuming that the model is given depth as input. Keeping the rest of the architecture unchanged, we train a pixel-aligned 3D Jacobian model. The model performs significantly worse numerically. Qualitatively, we found that this model fails to disentangle the sensitivities of surface points to different actuators, predicting incorrect Jacobians (**red boxes**) that over-explain viewpoints with occlusion. We hypothesize that this is due to a lack of multi-view supervision. For each robot, we show one successful example from both models, followed by two examples highlighting the baseline’s failure modes.